# Understanding and Mitigating Hallucination in Large Vision-Language Models via Modular Attribution and Intervention

**Tianyun Yang**[1,2]    **Ziniu Li**[3]    **Juan Cao**[1,2*]    **Chang Xu**[4*]

[1] Institute of Computing Technology, Chinese Academy of Sciences, Beijing, China

[2] University of Chinese Academy of Sciences, Beijing, China

[3] School of Data Science, The Chinese University of Hong Kong, Shenzhen, China

[4] School of Computer Science, Faculty of Engineering, University of Sydney, Australia

`yangtianyun19z@ict.ac.cn, ziniuli@link.cuhk.edu.cn`
`caojuan@ict.ac.cn, c.xu@sydney.edu.au`

## Abstract

Large Vision-Language Models (LVLMs) exhibit impressive capabilities in complex visual tasks but are prone to hallucination, especially in open-ended generation tasks. This paper explores why LVLMs tend to hallucinate and how to mitigate it. First, we conduct causal mediation analysis through counterfactual edits on specific modules in LVLMs. Our results disclose that Multi-Head Attention (MHA) modules contribute more to the probability of generating hallucination words than multi-layer perceptron modules. We then identify specific heads that are responsible for hallucination, referred to as hallucination heads. Second, we examine the behavior of hallucination heads. We find that they are concentrated in the middle and deeper layers, displaying a strong attention bias toward text tokens. Further, we show that the attention patterns of certain hallucination heads exhibit greater similarity to the base language model and change slowly during the instruction tuning process. Finally, we propose two simple yet effective methods to mitigate hallucination: one is training-free and can be applied directly during decoding, while the other involves fine-tuning. Both methods are *targeted* for hallucination heads to reduce their reliance on text tokens. Notably, our methods achieve up to 1.7x reduction in hallucination rate for the LLaVA-v1.5-7B model in COCO captioning task, outperforming existing baselines. Overall, our findings suggest that hallucinations in LVLMs are likely to stem from certain modules, and targeted interventions can effectively mitigate these issues.[1]

## 1 Introduction

Large Vision-Language Models (LVLMs) (Liu et al., 2024; Zhu et al., 2023; Tong et al., 2024; Li et al., 2023a; Bai et al., 2023) have demonstrated impressive capabilities in open-world visual tasks, ranging from basic perceptual tasks such as image description to complex reasoning tasks. Despite their strong capabilities, LVLMs often hallucinate, generating content that deviates from the provided image information (Bai et al., 2024), especially in long-form open-ended tasks like captioning. This reveals a fundamental weakness in current models and poses a risk of misleading users and undermining trust. Understanding the causes of hallucination in LVLMs and developing strategies to mitigate this issue is crucial for improving their reliability.

It is challenging to trace the causes of hallucinations. This complexity arises from the intricate training pipeline of LVLMs, which involves visual pre-training and instruction tuning built on top of a a pre-trained language model (Liu et al., 2024; Zhu et al., 2023). Additionally, the underlying architectures, often based on Transformers (Vaswani et al., 2017), are inherently complex, making interpretation difficult. Several studies have explored different factors contributing to hallucinations. Research by Leng et al. (2024) demonstrated that visual uncertainty from distorted input images can

---

*Corresponding anthor

[1]Code is available at `https://github.com/TianyunYoung/Hallucination-Attribution`

amplify hallucinations, while Huang et al. (2024) suggested that hallucinations often arise when the model overly relies on certain summary tokens during text generation, neglecting vital image information. Additionally, Li et al. (2023b) and Zhou et al. (2024) found that models tend to hallucinate by generating objects that frequently appear or co-occur with those in the images, leading to their erroneous inclusion in generated descriptions.

In this paper, we investigate the causes of hallucinations in LVLMs through the lens of attribution and intervention (Bach et al., 2015; Voita et al., 2019; Covert et al., 2021; Shah et al., 2024). Our goal is to uncover how the internal components of LVLMs contribute to hallucination words. Our study has three stages, which we will elaborate on below. Before diving in, we note that our primary focus is on the LLaVA-v1.5-7B model (LLaVA-7B for short) for open-ended generation tasks, with a particular focus on object hallucination. But we also investigate the LVLM model MiniGPT-4 (Zhu et al., 2023) in the Appendix to ensure our findings generalize.

First, we identify components responsible for generating hallucination words using counterfactual edits (zero-ablating specific modules), a tool widely used in mechanism interpretability (Nanda et al., 2023). We find that Multi-Head Attention (MHA) modules have a greater impact than multi-layer perceptron modules in producing hallucination words. Further, we localize this effect to specific heads, classifying them as hallucination or non-hallucination heads. We observe that hallucination heads mainly distribute in the middle and deeper layers of the Transformer.

Second, we examine how the identified hallucination heads contribute to hallucination, with two key insights. For one thing, hallucination heads favor textual inputs over visual inputs by up to five times. For another thing, the attention patterns of hallucination heads show a greater similarity to the base language model compared to non-hallucination heads. Furthermore, extensive instruction tuning (e.g., 665k samples in LLaVA-7B) results in limited changes to the attention patterns of prominent hallucination heads, suggesting that they are "lazy" when LVLMs are tuned with full parameters.

Finally, we develop methods to mitigate hallucination. Our preliminary results show that downscaling attention weights for text tokens significantly reduces the hallucination rate, while increasing attention on image tokens has no effect. Importantly, this intervention is effective specifically for hallucination heads. Based on these findings, we develop two approaches: one is training-free for use during decoding, while the other involves fine-tuning hallucination heads. Both methods reduce hallucination heads' reliance on text tokens and achieve up to 1.7x reduction in hallucination rate for the LLaVA-7B model on the COCO captioning task, outperforming existing baselines.

To summarize, our contributions have three fold:

- Attribution of Hallucination Components: We systematically identify and localize the components most responsible for hallucination generation in LVLMs. Specifically, we show that MHA modules, particularly certain heads in the middle and deeper layers, are key contributors.

- Analysis of Attention Bias: We show that hallucination heads strongly favor previously generated text over visual inputs. We also reveal that this pattern is inherited from the base language model and changes slowly during the visual instruction tuning process.

- Hallucination Mitigation Techniques: We develop two targeted strategies: one training-free for decoding and one involving fine-tuning. Both methods reduce over-reliance on text tokens, achieving a significant reduction in hallucination rates, outperforming existing baselines.

## 2 RELATED WORK

**Mitigating Hallucinations in LVLMs.** LVLMs' hallcunation behaviors are particularly severe in open-ended generation tasks (Huang et al., 2024; Zhou et al., 2024; Zhang et al., 2023). Many approaches have been explored to mitigate hallucinations in LVLMs, most of which focus on better decoding strategies. For instance, Leng et al. (2024) introduced visual contrastive decoding, which compares output distributions from original and distorted visual inputs to correct the model's over-reliance on unimodal priors and statistical bias. Moreover, Huang et al. (2024) observed that LVLMs frequently depend on the so-called summary tokens and proposed a method combining beam-search with retrospection-allocation, penalizing over-reliance on these tokens. Additionally, Chen et al. (2024) highlighted the importance of incorporating both local and global visual context, with the HALC method using an external grounding module during decoding. Furthermore, Zhou et al. (2024) developed LURE, which rectifies text by revising generated content to mitigate

hallucination issues like co-occurrence errors and object ambiguity. Deng et al. (2024) proposed a CLIP-guided decoding method to mitigate hallucination. Liu et al. (2023) tackled the issue by creating the LRV-Instruction dataset, featuring positive and negative instructions to improve robustness in visual instruction tuning. Zhang et al. (2024) introduced REVERIE, a large-scale dataset with reflective rational annotations, enabling models to justify response correctness. In contrast to previous studies, our work identifies a module-level cause of hallucination in LVLMs and develops targeted intervention strategies to mitigate hallucination effectively. In addition, a key feature of our method is that it requires only a single generation forward process during decoding, which is faster than existing methods.

**Interpretability of Transformers.** Understanding and explaining neural networks, particularly Transformers (Vaswani et al., 2017), is crucial for identifying their behaviors and limitations (Zhao et al., 2024). A widely used approach is causal mediation analysis (Hicks & Tingley, 2011), which attributes the contributions of key components, often employing "knock-out" techniques (Wang et al., 2023) to assess the impact of removing specific model elements on the output. Previous research (Voita et al., 2019; Olsson et al., 2022; Yu et al., 2023; Gandelsman et al., 2023) has demonstrated that individual attention heads in Transformers frequently assume distinct roles, such as induction, copying, and memorization. While prior studies, such as (Zhou et al., 2024), have also explored to understand hallucination in LVLMs, our work approaches it through the attribution of key components and intervenes in them specifically.

## 3 PRELIMINARY

LVLMs usually process both visual and linguistic data using three components: a vision encoder, a connector, and a Large Language Model (LLM). The vision encoder processes visual input, the connector aligns it with text tokens, and the LLM generates responses from this multimodal input. The LLM, structured as a transformer (Vaswani et al., 2017), consists of $L$ layers. Each layer includes a Multi-Head Attention (MHA) module and a Multi-Layer Perceptron (MLP), applying two primary residual transformations to the output of the previous layer $Z^{\ell-1}$:

$$\widehat{Z}^\ell = \text{MHA}^\ell(Z^{\ell-1}) + Z^{\ell-1}, \quad Z^\ell = \text{MLP}^\ell(\widehat{Z}^\ell) + \widehat{Z}^\ell.$$

In this framework, a layer $\ell$ employs an MHA module consisting of $H$ attention heads. Each head executes a self-attention operation, where the attention score is computed using query, key, and value matrices derived from the input. Specifically, for the $i$-th head in layer $\ell$, the operation is given by:

$$\text{head}_i^\ell(Z^{\ell-1}) = \text{Attention}(Q_i^\ell, K_i^\ell, V_i^\ell) = \text{softmax}\left(\frac{Q_i^\ell(K_i^\ell)^\top}{\sqrt{d_k}}\right)V_i^\ell,$$

where $Q_i^\ell = Z^{\ell-1}W_i^{Q,\ell}$ is the query matrix for the $i$-th head, and $K_i^\ell = Z^{\ell-1}W_i^{K,\ell}$ is the key matrix for the $i$-th head, and $V_i^\ell = Z^{\ell-1}W_i^{V,\ell}$ is the value matrix for the $i$-th head, and $d_k$ is the dimensionality of the key vectors. The outputs from all $H$ heads are then concatenated and projected using an output projection matrix $W^{O,\ell}$:

$$\text{MHA}^\ell(Z^{\ell-1}) = \text{Concat}(\text{head}_1^\ell(Z^{\ell-1}), \text{head}_2^\ell(Z^{\ell-1}), \ldots, \text{head}_H^\ell(Z^{\ell-1}))W^{O,\ell}.$$

For common 7B models such as LLaVA (Liu et al., 2024) and MiniGPT4 (Zhu et al., 2023), we have $L = 32$ and $H = 32$ and $d_k = 128$.

## 4 HALLUCINATION ATTRIBUTION AND INTERVENTION

In this paper, we diagnose hallucination behaviors in LVLMs by examining their internal components. We first identify the components most responsible for generating hallucination words in Section 4.1. Next, we analyze the patterns of these problematic components in Section 4.2. Finally, we propose strategies to mitigate hallucinations in Section 4.3.

### 4.1 TRACING HALLUCINATION BEHAVIORS TO MODEL COMPONENTS

The first step in our workflow is to select a neural network with notable hallucination behaviors for analysis. For this purpose, we choose the well-known LLaVA-7B.[2] We then break the neural network into smaller units for detailed investigation, focusing specifically on the MHA and MLP. We

---

[2]Our analysis and conclusions also hold for other models such as MiniGPT4; see Appendix A.1.

leverage causal mediation analysis (Hicks & Tingley, 2011). Specifically, we employ a "knockout" technique (Wang et al., 2023), selectively disabling the function of each component while maintaining the rest of the model intact. This is achieved through zero ablation[3], where the output of a targeted component is set to zero. Let $M$ be a vision-language model, and $M \setminus c$ be the counterpart of knock-outing of a component $c$. For a generated response, each component's influence score on hallucination is quantified using the following formula :

$$\mathcal{I}_c = \frac{1}{m} \sum_{t=1}^{T} \mathbb{I}\{y_t \in \text{hallucination}\} \left[ \mathbb{P}_M(y_t|v, x, y_{<t}) - \mathbb{P}_{M \setminus c}(y_t|v, x, y_{<t}) \right] \tag{1}$$

where $m = \sum_{t=1}^{T} \mathbb{I}\{y_t \in \text{hallucination}\}$ is the total number of hallucination words, $y_t$ is the token at time step $t$ in a generated response $y_{1:T}$, and $y_{<t}$ refers to the sequence of generated tokens prior to time step $t$. The variables $v$ and $x$ correspond to the visual context and user query, respectively. A high value $\mathcal{I}_c$ means that component $c$ contribute a lot to hallucination generation.

For our analysis, we use the COCO dataset (Lin et al., 2014), sampling 1,500 images from the training set. Following standard practice (Huang et al., 2024), we prompt the LVLM with the instruction:"Please describe the image in detail." Objects that match the ground truth labels are marked as non-hallucinated, while mismatches are classified as hallucinated. Subsequently, we perform zero-ablation on each MLP and MHA independently, calculate the influence score as in Equation (1), and present the average results for all MLPs and MHAs in Figure 1.

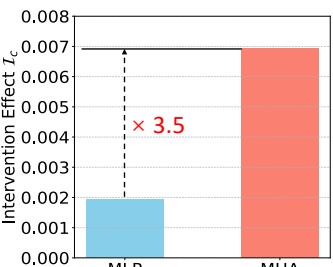

**MHAs have greater effects than MLPs for Hallucination.** We observe that removing MHAs has a significantly larger impact compared to removing MLPs, even though MHAs account for only half of the MLP parameters in LLaVA-7B. This finding aligns with (Gandelsman et al., 2023), which demonstrated that MHAs have a significant effect on classification accuracy in the transformer model CLIP (Radford et al., 2021). Intuitively, MHAs focus on capturing relationships and dependencies be-

Figure 1: Influence scores of MLP and MHAs in LLaVa-7B on generation probability of hallucination objects.

tween tokens through attention weights, whereas MLPs primarily process the information output by MHAs. The differing effects of these interventions suggest that hallucinations often stem from the model's attention to specific patterns or biases, highlighting the need for a more targeted analysis of these components.

Building on the above findings, we seek to further explore the behaviors of attention heads, especially those linked to hallucinations. The first step is identifying such attention heads. This task is not easy, as attention heads in Transformers fulfill many roles (Voita et al., 2019; Olsson et al., 2022) and often exhibit *polysemantic* functionality (Nanda & Sharkey, 2023; Bills et al., 2023), meaning they can influence multiple behaviors simultaneously. This makes it difficult to precisely isolate and identify the specific functions of each head. However, because our focus is on the generation of hallucination words, it is crucial to minimize the influence of non-hallucination words when attributing attention heads. To achieve this, we introduce a new criterion called **contrastive influence score**, which measures the difference between the intervention effect on hallucination words and its effect on non-hallucination words:[4]

$$\mathcal{I}_{h,\text{contrastive}} = \mathcal{I}_{h,\text{hallucination}} - \mathcal{I}_{h,\text{non-hallucination}}. \tag{2}$$

where $\mathcal{I}_{h,\text{hallucination}}$ is the intervention effect of attention head $h$ on generating hallucination words (see Equation (1)), while $\mathcal{I}_{h,\text{non-hallucination}}$ is the counterpart for non-hallucination worlds. Using the same model and setup as before, we calculate the contrastive influence values for a total of 1,024 attention heads across 32 layers, with 32 heads in each layer. The results are shown in Figure 2.

**Hallucination Heads Distribute in Middle and Last Layers.** For clarity, we categorize attention heads into two groups based on their contrastive influence scores: hallucination heads, which exhibit

---

[3]The technique of mean-ablating can also be used and similar results are observed; see Appendix A.1.

[4]It is important to use the contrastive influence score for diagnosing attention heads, as relying solely on $\mathcal{I}_{h,\text{hallucination}}$ for attribution and intervention does not work well in practice; see Appendix A.1.

high contrastive influence score, and non-hallucination heads, which show low contrastive influence score. Rather than using a strict threshold to define these categories, we apply a top-k selection, focusing on heads with the highest and lowest contrastive influence scores. Notably, both hallucination and non-hallucination heads, particularly the most prominent ones (e.g., the top 20 highlighted in boxes in Figure 2), are predominantly located in the middle and deeper layers of the model. This finding aligns with previous studies on Transformer models (Voita et al., 2019), which have shown that deeper layers tend to capture more abstract and task-specific representations.

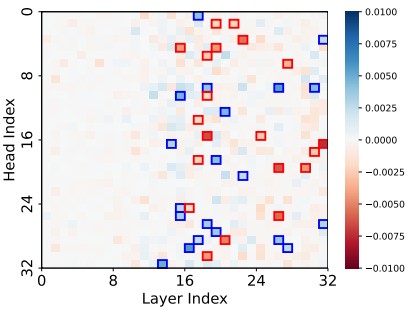

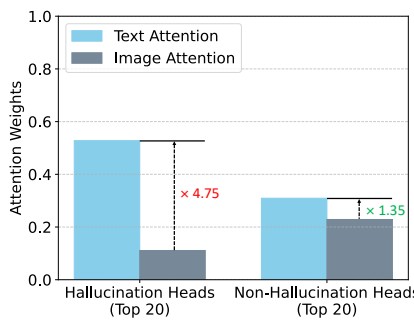

Figure 2: Contrastive influence values of attention heads in the LLaVA-7B model, with blue boxes for the top 20 hallucination heads and red boxes for non-hallucination heads.

Figure 3: Attention weights on text and image tokens for hallucination and non-hallucination heads. Hallucination heads strongly favor text tokens over image tokens.

## 4.2 BEHAVIOR ANALYSIS OF HALLUCINATION HEADS

After attributing attention heads responsible for hallucination in Section 4.1, we analyze their behavior patterns in this section. To understand why they induce hallucinations, we examine their attention maps. In particular, we divide the attention weights of an attention head into two parts: `text attention` and `image attention`. For each attention head, `text attention` is calculated by summing the attention weights assigned to tokens corresponding to instructions and responses[5], while `image attention` is determined by summing the attention weights assigned to tokens representing image features. See Figure 3 for the results.

**Hallucination Heads Favor Texts Over Image Inputs.** We observe that for hallucination heads, `text attention` is 4.75 times higher than `image attention`. In contrast, non-hallucination heads allocate attention more evenly between text and images tokens. This suggests that hallucination heads primarily focus on contextual text, causing LVLMs to rely on internal knowledge rather than image inputs when generating relevant words. This behavior helps explain the tendency toward hallucination. Our finding aligns with observations from previous research (Leng et al., 2024; Huang et al., 2024), which also suggest that LVLMs tend to overlook visual information during generation. However, a key difference is that we show this overlooking of visual information primarily occurs in hallucination heads rather than across all attention heads. This insight offers actionable guidance for developing targeted strategies to mitigate hallucinations, as discussed in Section 4.3.

So far, our analysis has focused on the model after visual instruction tuning. Now, we aim to take a step further and to investigate why hallucination heads over rely on text tokens in the tuned model. This is a challenging question, as factors like the training dataset, algorithm, and language model all determine the emergence of hallucination heads. Below, we attempt to link the attention patterns of hallucination heads in LLaVA-7B to those in its base language model, Vicuna-7B. We hypothesize that the over-reliance on text tokens in hallucination heads originates from the language model itself. To test this, we examine the attention map of Vicuna-7B, using the same question and response. Since Vicuna-7B cannot process image inputs directly, we use a placeholder token `<image>` to ensure contextual consistency. We then analyze the attention patterns within the text tokens (i.e., the responses). The results, displayed in Figure 4, show the similarity of attention patterns measured using cosine similarity.

---

[5]System tokens are excluded in the calculation here, as they often serve as "attention sink" and lack specific semantic meanings (Xiao et al., 2023). Thus, the sum of attentions weights in Figure 3 may not be 1.

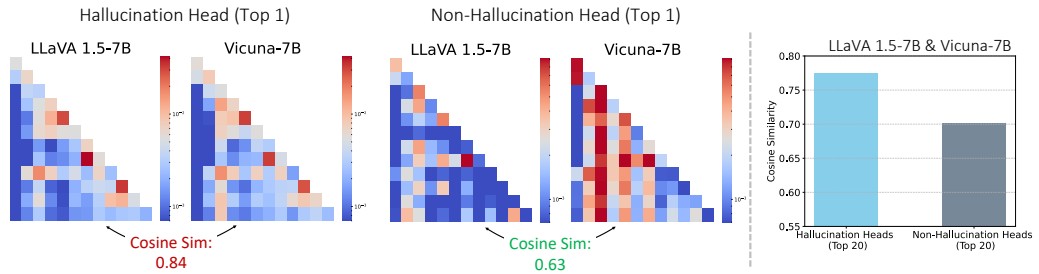

Figure 4: (Left and Middle): Attention maps of the top-1 hallucination heads and non-hallucination heads on generated text tokens of LVLM and its base LLM. Attention maps are downsampled for better visualization. (Right): Statistics over the top-20 attention heads.

**Inherited Attention Patterns in Hallucination Heads from Base Language Models.** We observe a notable similarity in the attention maps on generated text tokens between the hallucination heads of LLaVA-7B and Vicuna-7B, despite Vicuna-7B not processing actual image inputs. In contrast, non-hallucination heads do not exhibit such a clear pattern, particularly in the most prominent hallucination and non-hallucination heads. This suggests that hallucination heads may inherit much of their behavior from the base language model's next-token prediction.

**Some Hallucination Heads Show Slow Changes in Attention Maps During Visual Instruction Tuning.** The above results imply that hallucination heads likely inherit much of their behavior from the base language model, despite undergoing extensive visual instruction tuning (e.g., 665k samples for LLaVA-7B). To provide further evidence for this, we replicate the visual instruction tuning process of LLaVA-7B and calculate the Jensen-Shannon (JS) divergence between the attention maps before and after tuning. As shown in Figure 5, we find that top hallucination heads are "lazy", displaying noticeably slower changes in attention maps compared to non-hallucination heads. This insight could be valuable for the future development of LVLMs and warrants further investigation. For our work, this finding motivates the design of targeted fine-tuning strategies, rather than full-parameter tuning, to mitigate hallucinations in the next section.

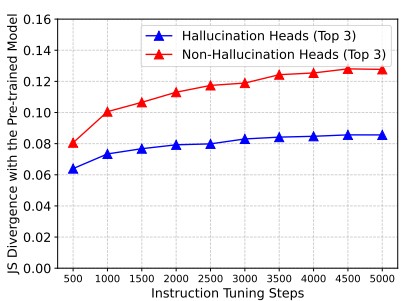

Figure 5: JS divergence of the attention map from the initial model (before visual instruction tuning) throughout the tuning process.

### 4.3 MITIGATING HALLUCINATION THROUGH MODULAR INTERVENTION

The findings in Section 4.2 lead us to explore a practical question: can hallucination be reduced if attention heads place less emphasis on text tokens, or alternatively, if they place more emphasis on image tokens? To investigate this, we adjust the text generation process by downscaling text attention and upscaling image attention through multiplication by a scaling factor. We randomly sample 500 COCO images from the validation set, and prompt the LVLM with the instruction: "Please describe the image in detail". The CHAIR metric is used to measure the hallucination rate by computing the proportion of objects mentioned in the generated description that are absent from the ground-truth labels. Furthermore, the BLEU score is used to assess the quality of the generated text. The results, as presented in Figure 6, offer three interesting insights.

- First, reducing text attention weights is more effective than increasing image attention weights.[6]

- Second, targeted intervention of text tokens on hallucination heads is more important than applying changes to the other attention heads.

---

[6]We provide an explanation Appendix A.2. We examine the linear spaces spanned by feature representations of text tokens and image tokens, respectively. We find that some directions in the text space cannot be represented by the image space, so changing image attention weights is not sufficient.

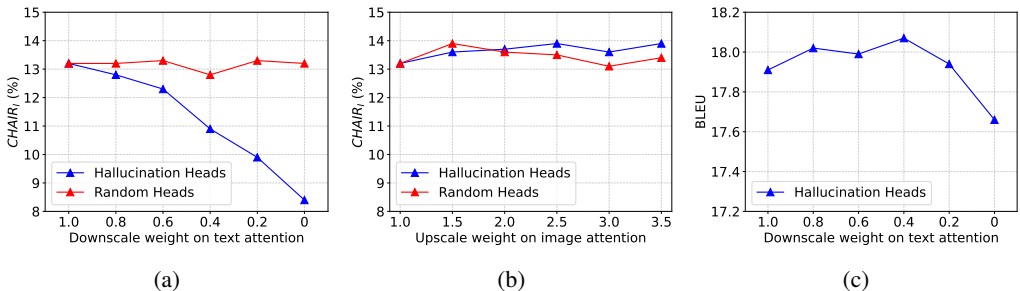

Figure 6: (a) Downscaling text attention weights reduce hallucination rate. (b) Upscaling image attention weights does not work. (c) Downscaling text attentions weights to zero lead to the drop in generation quality of BLEU.

- Last, simply downscaling text attention on hallucination heads to zero could hurt generation quality, as reflected in the BLEU score drops in Figure 6 (c).

Based on these insights, we propose two strategies to mitigate hallucination: adaptively deactivate text attention weights during the decoding stage (Section 4.3.1) and fine-tuning hallucination heads to specifically correct their attention patterns (Section 4.3.2).

### 4.3.1 ADAPTIVE DEACTIVATION OF HALLUCINATION HEADS

According to results presented in Figure 6, pruning the text attention weights of hallucination heads proves to be an effective decoding-time strategy but at the cost of the quality of text generation, potentially leading to less coherent outputs. To address this, we propose a more adaptive strategy, to deactivate text attention only for those hallucination heads that demonstrate excessive reliance on text attention during the decoding phase.

The adaptive deactivation mechanism operates as follows: at each decoding step $t$, for each hallucination head $h \in \mathcal{H}_{\text{hallucination}}$, we first compute the causal self-attention weights $A_h$ by applying the softmax function to the dot product of the query and key matrices for that head. To evaluate whether the head over-relies on text information in the current decoding step, we calculate an indicator value $I_h^{\text{text}}$ that accumulate the attention weights allocated to text tokens. If this indicator $I_h^{\text{text}}$ exceeds a predefined threshold $\tau$, we deactivate the head $H_i$ for that particular decoding step, i.e., setting the text attention weights to zero. Otherwise, the head remains active. See Algorithm 1.

---

**Algorithm 1** Adaptive Deactivation of Hallucination Heads (AD-HH)

**Require:** Hallucination Head Set $\mathcal{H}_{\text{hallucination}}$,
        Threshold $\tau$
1: **for** decoding time step $t$ **do**
2:    **if** attention head $h$ in $\mathcal{H}_{\text{hallucination}}$ **then**
3:       $I_h^{\text{text}} \leftarrow$ sum of text attention weights
4:       **if** $I_h^{\text{text}} > \tau$ **then**
5:          Set the text attention weights to zero
6:       Self-attention calculation
7:    **else**
8:       Self-attention calculation

---

**Algorithm 2** Targeted Fine-Tuning of Hallucination Heads (TF-HH)

**Require:** Hallucination Head Set $\mathcal{H}_{\text{hallucination}}$,
        dataset $\mathcal{D}$
1: **for** component $c$ **do**
2:    **if** $c$ in $\mathcal{H}_{\text{hallucination}}$ or $c$ is language head **then**
3:       c.requires_grad = true
4:    **else**
5:       c.requires_grad = false
6: **for** fine-tuning steps **do**
7:    Calculate the loss in Equation (3) for samples in $\mathcal{D}$
8:    Perform gradient descent update

---

We would like to note that Algorithm 1 requires a single generation forward process in the decoding stage. This differs from contrastive decoding methods (Chuang et al., 2024; Leng et al., 2024), which require two passes, or methods (Huang et al., 2024; Zhou et al., 2024) that rely on beam search and retrospection. As a result, Algorithm 1 runs faster in practice compared to these baselines. However, a key limitation of Algorithm 1 is that it requires the explicit calculation of attention weights, making it incompatible with memory-efficient mechanisms like FlashAttention (Dao, 2023). We address this issue by introducing another fine-tuning method below.

### 4.3.2 Targeted Fine-tuning of Hallucination Heads

In this section, we propose a targeted fine-tuning method (Algorithm 2) to mitigate hallucination. This approach directly addresses hallucination-prone attention heads in LVLMs and ensures strong performance with greedy decoding during the generation process. Given the desired goals mentioned in the beginning of Section 4.3, we propose a simple training objective that combines the standard next-token-prediction objective for maintaining generation quality and a text-attention penalty for reducing hallucinations. For a particular training sample $(v, x, y_{1:T})$, where $v$, $x$, and $y_{1:T}$ denote the visual input, textual instruction and response, we define the loss function:

$$\mathcal{L}(v, x, y_{1:T}) = \sum_{t=1}^{T} \left[ \underbrace{-\log P_M(y_t | v, x, y_{<t})}_{\text{next-token-prediction}} + \lambda \sum_{h \in \mathcal{H}_{\text{hallucination}}} \underbrace{\log A_h^{\text{text}}(v, x, y_{<t})}_{\text{text-reliance-reduction}} \right], \quad (3)$$

where $\lambda > 0$ controls the strength of the penalize for text reliance, and $A_h^{\text{text}}$ is the summed attention weights on text tokens of hallucination head $h$. A key design consideration in our method is the choice of the optimization variables. Following the guidance from previous results, we only fine-tune the hallucination heads rather than full heads. Additionally, we fine-tune the final layer of the language model's prediction head (referred to as the language head in Algorithm 2) to enhance the model's capacity in fine-tuning. Because our fine-tuning method is targeted, it is highly effective in correcting hallucination heads. A small number of optimization steps is sufficient to achieve significant improvements. In practice, we found that 200 optimization steps for LLaVA-7B yielded good results, requiring less than 3% of the compute used for instruction tuning in LLaVA-7B.

## 5 Experiments

### 5.1 Experiment Setup

In addition to the previously examined LLaVA-7B model, we also investigate the well-known LVLM model MiniGPT-4 (Zhu et al., 2023), which has 7B parameters as LLaVA.

**Proposed Methods.** For the decoding-time method AD-HH in Algorithm 1, we select the top 20 attention heads for LLaVA-7B and the top 10 heads for MiniGPT-4. We use fewer heads for MiniGPT-4 because fewer heads are remarkable in the contrastive influence map (see Appendix A.1). For the threshold $\tau$ to control the when to deactivate text attention in decoding, we find that setting $\tau$ around 0.5 provides a good balance between generation quality and hallucination reduction. Specifically, we set $\tau = 0.4$ for LLaVA-7B and $\tau = 0.5$ for MiniGPT-4. For the fine-tuning method TF-HH in Algorithm 2, we slightly increase the number of attention heads to 30 and 20, to increase the representation power. For the penalty weight $\lambda$ in fine-tuning, we set it as 2. Both of our methods are based on greedy decoding to generate samples for evaluation.

**Baselines.** Alongside proposed methods for mitigating hallucination, we also study baseline approaches from prior literature, including the standard greedy decoding method and several state-of-the-art techniques: OPERA (Huang et al., 2024), VCD (Leng et al., 2024), LURE (Zhou et al., 2024), and HALC (Chen et al., 2024). Additionally, we include DoLA (Chuang et al., 2024), which was originally designed to enhance factuality in language models and has also been studied in previous literature. Hyper-parameters of these baselines follow from previous literature and are provided in Appendix B for reference.

**Datasets.** We mainly focus on hallucination in open-ended generation tasks to assess the effectiveness of our methods. To evaluate object hallucination in visual caption, we use images from the COCO validation and Nocaps (Agrawal et al., 2019) datasets. For Nocaps, we use the out-of-domain version. We randomly select 500 samples from each and prompt the LVLMs with the query, "Please describe this image in detail". We also examine the performance of LVLMs on complex tasks using the MM-Vet (Yu et al., 2024) dataset, which assesses integrations of six core vision-language capabilities: recognition, OCR, knowledge, language generation, spatial awareness, and math. The maximum token limit is set to 128 by default.

**Evaluation Metrics.** To evaluate object hallucination in image caption, we employ CHAIR metrics (Rohrbach et al., 2018), designed for automatic hallucination assessment. CHAIR measures the hallucination rate by computing the proportion of objects mentioned in a generated description that are absent from the ground-truth labels. The metric is split into two components: sentence-level

Table 1: Hallucination rates in terms of CHAIR$_S$ and CHAIR$_I$ on COCO and Nocaps (Out-of-Domain) image captioning tasks, with lower value indicating better performance. The subscript † indicates that this method is training-free, while ∗ means that this method is training-based.

| Methods | COCO | | | | Nocaps (Out-of-Domain) | | | |
|---|---|---|---|---|---|---|---|---|
| | LLaVA-7B | | MiniGPT-4 | | LLaVA-7B | | MiniGPT-4 | |
| | CHAIR$_S$ | CHAIR$_I$ | CHAIR$_S$ | CHAIR$_I$ | CHAIR$_S$ | CHAIR$_I$ | CHAIR$_S$ | CHAIR$_I$ |
| Greedy† | 51.8 | 13.3 | 40.6 | 13.7 | 43.2 | 14.3 | 57.4 | 20.0 |
| DoLA† | 53.8 | 13.9 | 41.0 | 13.8 | 42.0 | 13.7 | 57.2 | 20.4 |
| OPERA† | 50.2 | 14.5 | 35.2 | 12.8 | 44.2 | 14.4 | 46.2 | 16.2 |
| VCD† | 55.4 | 15.7 | 38.8 | 14.8 | 43.6 | 14.4 | 48.2 | 17.5 |
| LURE† | 51.2 | 13.4 | 46.4 | 14.2 | 41.8 | 14.4 | 55.8 | 19.6 |
| HALC† | 50.2 | 12.4 | 36.4 | 11.8 | 40.2 | 12.2 | 53.0 | 18.0 |
| AD-HH† (Ours) | **29.6** | **8.0** | 35.2 | 11.7 | 35.6 | **9.4** | 46.8 | **16.2** |
| TF-HH∗ (Ours) | 35.0 | 8.7 | **32.0** | **11.4** | **35.4** | 11.1 | **45.2** | 16.8 |

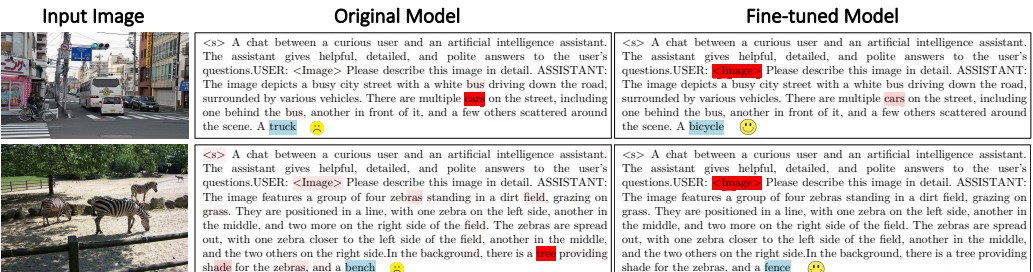

Figure 7: Visualization of attention weights for the top hallucination head when predicting the next token (shown in blue). We accumulate the attention weights on a total of 576 image tokens into the placeholder `<image>` to simplify visualization. Redder values in the context indicate larger attention weights. The original LLaVA-7B model significantly relies on the previously generated tokens, resulting in hallucination. Our model through targeted fine-tuning does not have this issue.

hallucination (CHAIR$_S$) and image-level hallucination (CHAIR$_I$). Specifically:

$$\text{CHAIR}_S = \frac{|\text{captions with hallucinated objects}|}{|\text{all captions}|}, \quad \text{CHAIR}_I = \frac{|\text{hallucinated objects}|}{|\text{all mentioned objects}|}. \quad (4)$$

A smaller value of CHAIR$_S$ and CHAIR$_I$ indicate a lower hallucination rate.

## 5.2 Main Results

**Modular Intervention Can Reduce Hallucination by up to 1.7x.** We present the evaluation results on COCO in Table 1 (left). Our methods—either decoding or fine-tuning—achieve consistent improvements for both models, reducing hallucination rates by up to 1.7 times compared to greedy decoding for LLaVA-7B and 1.3 times for MiniGPT-4. We also note that reducing hallucinations does not compromise text generation quality, which is shown in Table 4 in the Appendix A.3. We find that DoLA is ineffective in this scenario, a finding also observed in the HALC paper. This is likely because DoLA is designed to elicit factual knowledge from the model, which may unintentionally amplify language biases inherited from the base language model, making it unsuitable for mitigating hallucinations in LVLMs.

To illustrate the effectiveness of our methods, we present case studies in Figure 7. The original model demonstrates a strong reliance on text tokens, which leads to the generation of hallucination objects that have close semantic relations with the focus token. We then provide the same prompt before the hallucinated token to our fine-tuned model. We can observe that the fine-tuned model shifts attention more towards the image tokens, resulting in the generation of image-consistent objects.

**Modular Attribution and Invention Show Transferbility.** Recall that the hallucination heads in our methods were identified using the COCO training dataset, and the above evaluation is with the COCO validation dataset. To examine the transferability and robustness across datasets, we evaluate

performance on the Nocaps (out-of-domain) dataset, which includes objects not present in COCO. We report the results in Table 1 (right). We observe similar conclusions: our method maintains strong performance on out-of-domain datasets, indicating that modifications to the hallucination heads have a broad impact across tasks and implying the generalizability of our method.

Table 2: Performance in complicated multimodal tasks from MM-Vet (Yu et al., 2024), with higher value indicating better performance.

| Methods | LLaVA-7B | | | | | | | MiniGPT-4 | | | | | | |
|---|---|---|---|---|---|---|---|---|---|---|---|---|---|---|
| | Rec | OCR | Know | Gen | Spat | Math | Total | Rec | OCR | Know | Gen | Spat | Math | Total |
| Greedy[†] | 36.3 | 21.8 | 17.0 | 18.9 | 24.8 | 7.7 | 31.4 | 26.5 | 13.3 | 17.5 | 13.9 | 22.3 | 8.1 | 22.2 |
| DoLA[†] | 37.2 | 22.1 | 17.9 | 21.0 | 26.3 | 7.7 | 31.7 | 24.9 | 12.9 | 18.5 | 12.0 | 21.7 | 7.7 | 21.6 |
| OPERA[†] | 35.4 | 25.6 | 20.5 | 22.9 | 30.9 | 11.5 | 32.0 | 28.2 | 15.0 | 16.5 | 11.4 | 21.9 | 11.5 | 23.6 |
| VCD[†] | 33.0 | 23.6 | 16.0 | 19.4 | 25.6 | 3.8 | 29.4 | 25.3 | 14.8 | 17.4 | 15.0 | 20.3 | 0.0 | 20.9 |
| HALC[†] | 36.2 | 21.5 | 17.5 | 20.1 | 23.5 | 7.7 | 30.8 | 24.9 | 15.7 | 15.2 | 10.7 | 23.2 | 7.7 | 21.7 |
| AD-HH (Ours)[†] | 38.4 | 26.0 | 21.2 | 21.9 | 30.3 | 7.7 | 34.3 | 28.2 | 16.6 | 16.1 | 13.7 | 26.1 | 12.0 | 23.8 |
| TF-HH (Ours)* | 36.6 | 24.1 | 17.9 | 19.0 | 27.2 | 11.5 | 32.5 | 31.9 | 18.1 | 22.3 | 16.6 | 26.5 | 18.5 | 27.3 |

**Modular Intervention Benefits Complicated Multimodal Tasks.** To further validate our method's effectiveness on complex open-ended multimodal tasks, we evaluated performance on the MM-Vet dataset, which assesses six multimodal capabilities: recognition, OCR, knowledge, language generation, spatial awareness, and math. Table 2 shows that our method, either through decoding or fine-tuning, also improves a range of multimodal capacibilites. For instance, our decoding method boosts LLaVA-7B's scores in OCR and spatial awareness by 4.2 and 5.5 points, respectively. Similarly, our fine-tuning method enhances MiniGPT-4's performance in recognition and math, with improvements of 5.4 and 10.4 points. These results demonstrate that modifying hallucination heads benefits not only tasks focused on hallucination reduction but also general multimodal tasks.

**Generation Time Comparison.** We compare the generation time of our decoding method AD-HH with existing decoding-time hallucination mitigation methods in Figure 8. For OPERA and our method that requires explicit attention weights, we use the standard self-attention implementation. For Greedy, DoLA, VCD, and HALC, where explicit attention weights are not needed, we employ Flash-Attention, which is generally faster than standard self-attention. All methods were tested on a single A100-80GB GPU. We observe that our method achieves similar decoding times to greedy decoding. This is because we intervene on the attention weights on-the-fly during the generation process. In contrast, other methods inevitably introduce computational overhead. For instance, VCD requires a double inference process for contrastive decoding, and OPERA requires retrospecting to previous steps when knowledge aggregation happens.

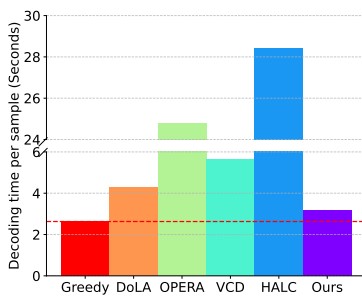

Figure 8: Generation time of a single response.

## 6  CONCLUSION

In this work, we explore why vision-language models generate hallucinated words by examining their internal structures. Our analysis reveals that multi-head attention modules are key contributors, with certain heads in the middle and deeper layers disproportionately focusing on text tokens over visual inputs. These attention patterns appear to be inherited from the base language model, with some hallucination heads showing slow adaptation during instruction tuning. To address this, we propose both decoding-time and fine-tuning strategies to mitigate hallucinations. Our modular approach offers a practical and efficient post-training solution for managing the behavior of LVLMs, in contrast to full-parameter tuning and intervention. This approach is especially valuable given the trend of increasing model sizes in LVLMs.

## 7 ACKNOWLEDGMENT

Tianyun Yang would like to thank Zeyu Qin and Yunke Wang for helpful discussion about the algorithm design. This work was supported in part by the Australian Research Council under Projects DP240101848 and FT230100549, China Scholarship Council, and Strategic Priority Research Program of the Chinese Academy of Sciences (No. XDB0680202).

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

## A    ADDITIONAL RESULTS

### A.1    ADDITIONAL RESULTS ON COMPONENT ATTRIBUTION

**On the importance of contrastive influence metric.** To validate the proposed contrastive influence metric, which isolates a head's effect on generating hallucinated versus non-hallucinated objects, we compare it to the non-contrastive influence, i.e., directly use $\mathcal{I}_{h,\text{hallucination}}$ for identifying hallucination heads. In Table 3, we report the evaluation results for LLaVA-7B on the COCO captioning task, where we adaptively deactivate the top-20 hallucination heads identified through contrastive and non-contrastive methods. As shown, deactivating heads identified by contrastive influence results in a significantly greater reduction in hallucinations, highlighting its superior precision in localizing hallucination heads.

Table 3: Ablation study on constrative influence.

| Method | CHAIR$_S$ | CHAIR$_I$ |
|---|---|---|
| Greedy | 51.8 | 13.3 |
| AD-HH (Non-contrastive Influence) | 41.8 | 11.0 |
| AD-HH (Constrative Influence) | **29.6** | **8.8** |

**Sensitivity analysis of the causal mediation method.** In Figure 9(a), we plot the Spearman rank correlation between the contrastive influence scores of each attention head found using $N$ samples and those using 1000 samples. Here, $N$ represents the number of samples used for hallucination head attribution. Varying $N$ from 50 to 1000, we observe that when $N$ reaches 500, the Spearman rank correlation is 0.93 compared to the results with $N = 1000$. Beyond $N = 500$, increasing the number of samples results in minimal changes to the attribution outcomes. This indicates that 500 samples are sufficient for accurately identifying hallucination heads.

We also evaluate alternative "knock-out" techniques for identifying hallucination heads, including: 1) using log probabilities instead of probabilities for intervention effects, and 2) replacing the output of the target component with the mean value of the hidden state outputs. In Figure 9(b,c), we plot the Pearson correlation between these two alternative methods and our default zero-ablation method using probabilities. The high correlation observed suggests that the intervention results (contrastive influence) of each head align closely with the default methods, indicating that our approach exhibits low sensitivity.

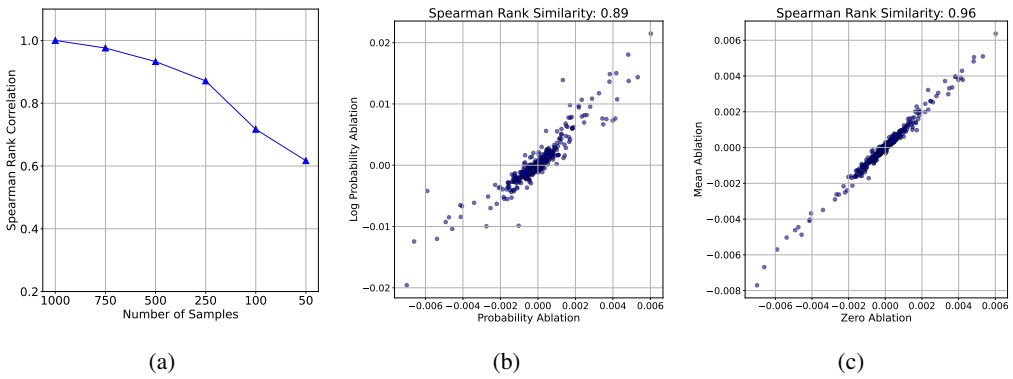

(a)                                    (b)                                    (c)

Figure 9: (a) Ablation study on number of samples to identify hallucination heads. (b) Spearman rank similarity comparison between effects calculated on log probability and probability. (c) Spearman rank similarity comparison between mean-ablation and zero-ablation methods.

**Component attribution results on MiniGPT-4.** In Figure 10, we present the constrastive influence of attention heads in the MiniGPT-4 model on generating hallucinated and non-hallucinated objects. As indicated in the Figure, the hallucination heads also are distributed in the latter half of the model.

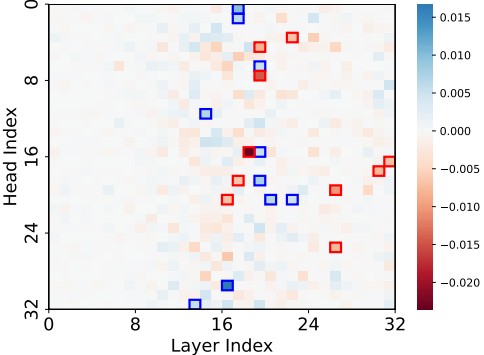 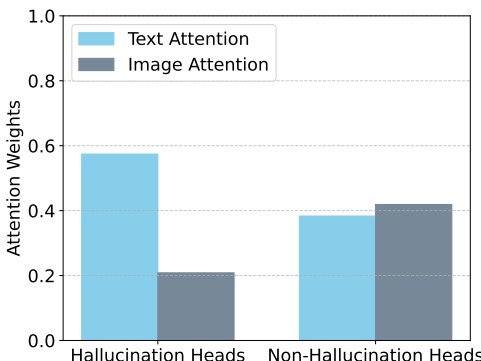

Figure 10: Contrastive influence of attention heads in the MiniGPT-4 model. Bluer boxes indicate heads more responsible for generating hallucinated objects.

Figure 11: Averaged attention weights on text and image tokens for hallucination and non-hallucination heads identified in MiniGPT-4.

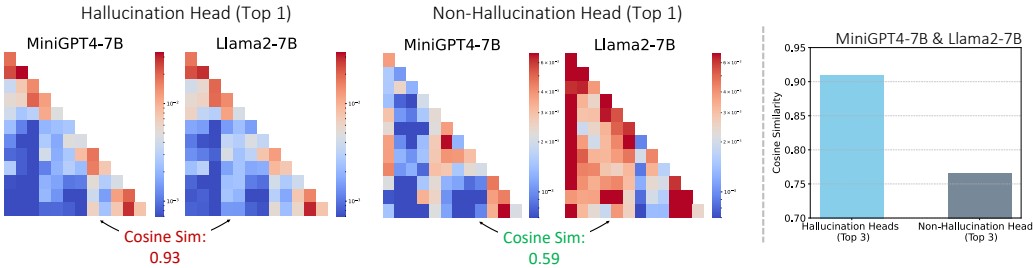

Figure 12: Hallucination heads in MiniGPT-4 inherit the text attention pattern from the base LLM.

## A.2 ADDITIONAL RESULTS ON BEHAVIOUR ANALYSIS OF HALLUCINATION HEADS

**Attention bias of hallucination heads in MiniGPT-4.** In Figure 11, we plot the averaged attention weights on text and image tokens for top-3 hallucination and non-hallucination heads. As shown in the Figure, the hallucination heads of MiniGPT-4 also demonstrate much stronger attention bias than non-hallucination heads.

**Inherited attention pattern of hallucination heads in MiniGPT-4.** In Figure 12, we compare the attention patterns of hallucination heads in MiniGPT-4 with those in its base language model, Llama-2-7B. As shown, the text attention patterns in MiniGPT-4 are more aligned with Llama-2-7B for hallucination heads than for non-hallucination heads. Specifically, the top hallucination head with the highest contrastive influence shows a cosine similarity as high as 0.93, in contrast to the top non-hallucination head, which only exhibits a similarity of 0.59.

**Spanned linear space analysis.** Our findings in Section 4.3 indicate that downscaling text attention weights is more effective than upscaling image attention weights. We hypothesize that this can be explained by the linear spaces spanned by text tokens and image tokens. If the space spanned by text tokens is significantly larger than that of image tokens, simply adjusting image attention weights (i.e., modifying the linear combination in the self-attention mechanism) may be insufficient.

To test this, we construct a linear space for text tokens, denoted as $\mathcal{S}_{\text{text}}$, and a linear space for image tokens, denoted as $\mathcal{S}_{\text{image}}$. We then calculate the projection distances:

$$d(\text{projection } \mathcal{S}_{\text{image}} \text{ onto } \mathcal{S}_{\text{text}}) = \|\mathcal{S}_{\text{text}}(\mathcal{S}_{\text{text}}^\top \mathcal{S}_{\text{text}})^{-1} \mathcal{S}_{\text{text}}^\top \mathcal{S}_{\text{image}}\|_F = 3636.3$$

$$d(\text{projection } \mathcal{S}_{\text{text}} \text{ onto } \mathcal{S}_{\text{image}}) = \|\mathcal{S}_{\text{image}}(\mathcal{S}_{\text{image}}^\top \mathcal{S}_{\text{image}})^{-1} \mathcal{S}_{\text{image}}^\top \mathcal{S}_{\text{text}}\|_F = 15688.5$$

The results suggest that certain directions in the text space cannot be linearly represented by the image token features. As a result, even with careful tuning of image attention weights, the output of the self-attention mechanism may still retain components of textual information, contributing to hallucination.

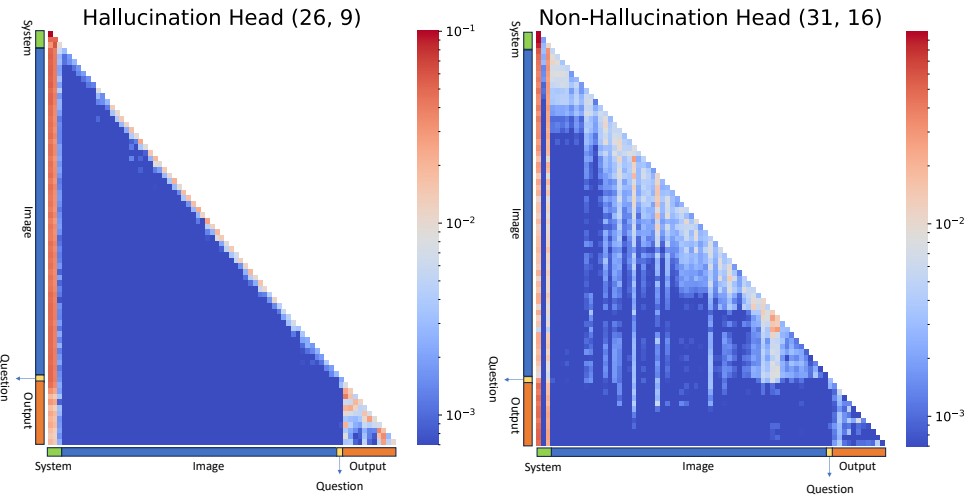

Figure 13: Complete attention maps of hallucination and non-hallucination heads, encompassing system, image, question, and output tokens. One cell in the maps denotes 10 tokens.

**Complete attention maps of hallucination and non-hallucination heads.** In Figure 13, we present the complete attention maps of two representative hallucination and non-hallucination heads on the COCO captioning task, encompassing system, image, question, and output tokens. For clarity, the maps are downscaled by a factor of 10. As illustrated in the figure, the hallucination head assigns significantly lower attention scores to image tokens, suggesting a tendency to overlook them. In contrast, the non-hallucination head demonstrates a more balanced attention distribution between image and text tokens.

Further, we would like to emphasize that this over-reliance is a symptom of hallucination heads, not a criterion to identify them. We also note that there may be other attention heads that also exhibit over-reliance on text tokens but do not influence hallucination behavior (e.g., layer 31, head 4 in LLaVA-7B). This occurs because some heads may function as a general-purpose language head, ensuring fluency and coherence in text generation. Although these heads look like the left attention map in Figure 13 (left) and over-rely on text tokens, they are unrelated to hallucination behavior. They qualify as hallucination heads only when they are causally responsible for hallucination behavior, indicated by large contrastive influence values.

## A.3 Additional Results on Modular Intervention

**On the effectiveness of adaptive deactivation.** To illustrate the effectiveness of adaptive deactivation of hallucination heads in preserving the generation quality and mitigating hallucination, we compare the performance of our proposed adaptive deactivation method and full deactivation, which completely setting the text attention weights of hallucination heads to zero regardless of the input. Figure 14 shows the the hallucination rate ($CHAIR_I$) and generation quality (BLEU) of the two methods as the number of top-$k$ hallucination heads to be deactivated varied. As illustrated in the Figure, adaptive deactivation yeilds more optimal hallucination reduction and generation quality maintaining with the same number of top-k hallucination heads to be deactivated. This indicates that context-aware pruning is more flexible than static pruning method.

**On the top-k hallucination head selection.** In Figure 15, we present the relationship between the hallucination rate ($CHAIR_I$) and generation quality (BLEU) as the number $k$ of deactivated hallucination heads increases. As illustrated, increasing $k$ leads to more effective hallucination reduction

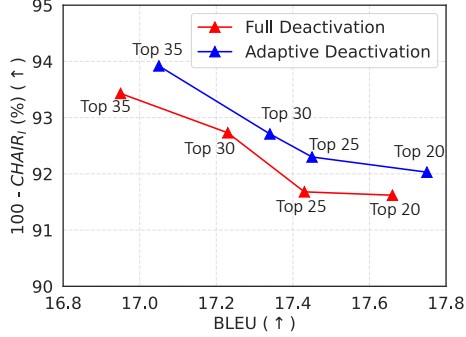

Figure 14: Compare adaptive deactivation and full deactivation in reducing hallucination and maintaining generation quality.

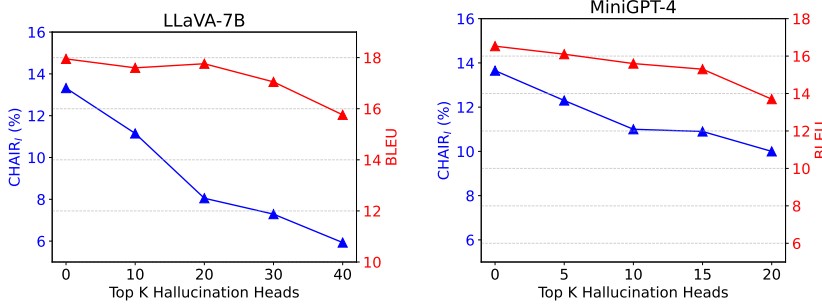

Figure 15: Ablation study on the top-k hallucination head selection.

Table 4: Generation quality comparison, with higher value indicating better performance.

| Dataset | COCO | | | Nocaps | | |
| --- | --- | --- | --- | --- | --- | --- |
| | BLEU | ROUGH | METEOR | BLEU | ROUGH | METEOR |
| Greedy | 17.9 | 18.8 | 18.2 | 24.6 | 21.8 | 18.8 |
| AD-HH (Ours) | 17.8 | 19.1 | 18.1 | 23.1 | 21.3 | 18.3 |
| TF-HH (Ours) | **18.8** | **20.0** | **18.7** | **25.5** | **22.9** | **19.2** |

but is also associated with a decline in generation quality after a certain number of deactivated hallucination heads. We find that for LLaVA-7B, $k = 20$ yields a favorable trade-off between generation quality and hallucination reduction for LLaVA-1.5, while $k = 10$ is optimal for MiniGPT4, which we adopt as our default parameter in adaptive deactivation of hallucination heads.

**On the importance of fine-tuning hallucination heads.** To highlight the importance of targeting only hallucination heads, we compare the results in Table 5 for fine-tuning the full parameters, 30 randomly selected heads, and the top 30 hallucination heads on LLaVA-1.5. As shown in the table, fine-tuning only the hallucination heads achieves significantly more hallucination reduction compared to both full fine-tuning and fine-tuning random heads. This verifies that hallucination is mainly caused by only a small portion (less than 3%) of attention heads, and focusing on them is crucial for reducing hallucination.

**On the generation quality comparison.** In Table 15 and Table 16, we visualize some image description examples of the LLaVA-7B model. The hallucinated objects are highlighted in RED. The main results show that our method does not influences the the coherent and fluency of generated context. We qualitatively measure the generation quality in Table 4. For BLEU, ROUGH (Lin & Och, 2004), and METEOR (Banerjee & Lavie, 2005), the adaptive deactivation method shows a slight decrease on the Nocaps dataset. While the fine-tuning method maintains and even improve the quality, which indicates that fine-tuning provides a more robust and safe adjustment to the model.

Table 5: Ablation study on fine-tuned components.

| Methods | CHAIR$_S$ | CHAIR$_I$ |
|---|---|---|
| Original Model | 51.8 | 13.3 |
| Fine-tune Full Parameters | 56.3 | 15.6 |
| Fine-tune Random Heads | 54.4 | 15.7 |
| Fine-tune Hallucination Heads | **35.0** | **8.7** |

Table 6: Human eval results on non-hallucination score and generation quality score. Higher values indicate lower hallucination rate and better quality.

| Methods | Non-Hallucination Score | Generation Quality Score |
|---|---|---|
| Greedy | 3.25 | 3.99 |
| AD-HH (Ours) | **3.87** | 3.85 |
| TF-HH (Ours) | 3.78 | **4.01** |

Table 7: Results on the MME dataset.

| Method | Score |
|---|---|
| Greedy | 1791.64 |
| AD-HH (Ours) | 1812.36 |
| TF-HH (Ours) | **1813.06** |

Table 8: Comparison results with additional baseline methods.

| | Greedy | GCD | LRV | REVERIE | AD-HH (Ours) | TF-HH (Ours) |
|---|---|---|---|---|---|---|
| CHAIR_S | 51.8 | 39.2 | 39.4 | 49.6 | **29.6** | 35.0 |
| CHAIR_I | 13.3 | 10.8 | 13.1 | 12.7 | **8.0** | 8.7 |

We also conduct a manual evaluation study in the help of two Ph.D. students and one undergraduate student to evaluate the responses. They were instructed to score each response on a scale of 1 to 5 based on two criteria: (1) non-hallucination performance, with higher scores reflecting fewer hallucinations, and (2) generation quality, with higher scores indicating more fluent and descriptive responses. For both the baseline and our proposed methods, the evaluators assessed a total of 500 generated responses per method, resulting in 1500 responses overall. The evaluation results are shown in Table 6. These results demonstrate that our methods effectively mitigate hallucination while maintaining high generation quality.

**Evaluation on the MME dataset.** We also evaluate on the MME benchmark Fu et al. (2023), which measures the perception and cognition abilities of LVLMs. As demonstrated in Table 7, our two methods show generalization ability on the benchmark, improving the overall score by about 20 absolute points.

**Comparison with additional baseline methods.** We extend our comparison by evaluating our method against additional baseline methods on the COCO dataset using the LLaVA-7B model, as presented in Table 8. The additional baseline methods include GCD (Deng et al., 2024), LRV (Liu et al., 2023), and REVERIE (Zhang et al., 2024). The results demonstrate that our approach, which leverages targeted interventions, is significantly more effective in mitigating object hallucinations in open-ended generation tasks.

Note that we re-implemented these baselines on our model and dataset to ensure a fair comparison, as there are differences in the models and evaluation datasets used. Specifically, the GCD paper evaluates on the COCO Karpathy Test set, whereas we use the COCO validation set. Additionally,

Table 9: Evaluation on larger and more recent LVLM models.

| Methods | Llama-3.2-11B-Vision | | LLaVA-v1.5-13B | | Chameleon-30B | | LLaVA-v1.6-34B | |
|---|---|---|---|---|---|---|---|---|
| | $CHAIR_S$ | $CHAIR_I$ | $CHAIR_S$ | $CHAIR_I$ | $CHAIR_S$ | $CHAIR_I$ | $CHAIR_S$ | $CHAIR_I$ |
| Greedy | 28.4 | 7.4 | 48.6 | 12.4 | 38.0 | 12.6 | 23.2 | 6.4 |
| AD-HH (Ours) | **22.6** | **4.9** | **38.8** | **9.4** | **34.8** | **12.5** | **20.4** | **5.6** |

Table 10: Number and ratio of salient hallucination heads across different scales of LVLMs.

| | LLaVA-7B | LLaVA-13B | LLaVA-34B |
|---|---|---|---|
| Number of Salient Hallucination Heads | 42 | 37 | 10 |
| Ratio of Salient Hallucination Heads | 4.1% | 2.3% | 0.3% |

the provided MiniGPT-4 checkpoint by REVERIE is pretrained on Llama-7B, while our experiments are based on Llama2-7B.

**Evaluation on larger and more recent LVLM models.** We extend experiments to larger and more recent models, including Llama-3.2-11B-Vision[7], LLaVA-v1.5-13B[8], Chameleon-30B (Team, 2024), and LLaVA-v1.6-34B[9]. These models are both modern and representative, with Llama-3.2-11B-Vision being released just two months ago, Chameleon pioneers an early-fusion-based multimodal training strategy, LLaVA-Next-34B increases the input image resolution to 4x more pixels to grasp more visual details. Using the same settings on the COCO dataset, our method consistent improvements, as indicated in Table 9. Our method reduces $CHAIR_S$ by approximately 6 points for Llama-3.2-11B-Vision, 10 points for LLaVA-v1.5-13B, 3.2 points for Chameleon-30B, and 2.8 points improvement for LLaVA-v1.6-34B.

**Hallucination behaviour across different scales of LVLMs.** Inspired by the reviewer's advice, we conducted an empirical analysis to investigate whether hallucination heads exist across different scales of LVLM models. Specifically, we examined various scales of LLaVA models, including LLaVA-7B, LLaVA-13B, and LLaVA-34B. Using the contrastive map derived from Equation 2 for these models, we identified salient hallucination heads whose contrastive influence values exceeding 25% of the maximum contrastive influence value. These heads are deemed most responsible for hallucination behaviors. We evaluate both their absolute numbers and their ratio relative to all attention heads. The results are presented in Table 10.

Our findings indicate that hallucination heads tend to diminish as model size increases and sufficient post-training is applied. While we cannot entirely disentangle the contributions of individual factors, such as the LLM backbone, data size, and data sources, or the image tokenizer, our observations align with our hypothesis: larger models exhibit stronger representational power to learn correct behaviors from data, whereas smaller models are more prone to language bias.

## B  EXPERIMENT DETAILS

Table 11: DoLA Hyperparameters.

| Parameters | Value |
|---|---|
| Adaptive Plausibility Threshold | 0.1 |
| Early Exist Layers | $[0, 2, 4, \cdots, 32]$ |

**Implementation of Baselines.** In Table 11, Table 12, Table 13, and Table 14, we present the hyperparameters for the baseline methods: DoLA, OPERA, VCD, and HALC. These parameters follow

---

[7]https://ai.meta.com/blog/llama-3-2-connect-2024-vision-edge-mobile-devices
[8]https://huggingface.co/liuhaotian/llava-v1.5-13b
[9]https://llava-vl.github.io/blog/2024-01-30-llava-next

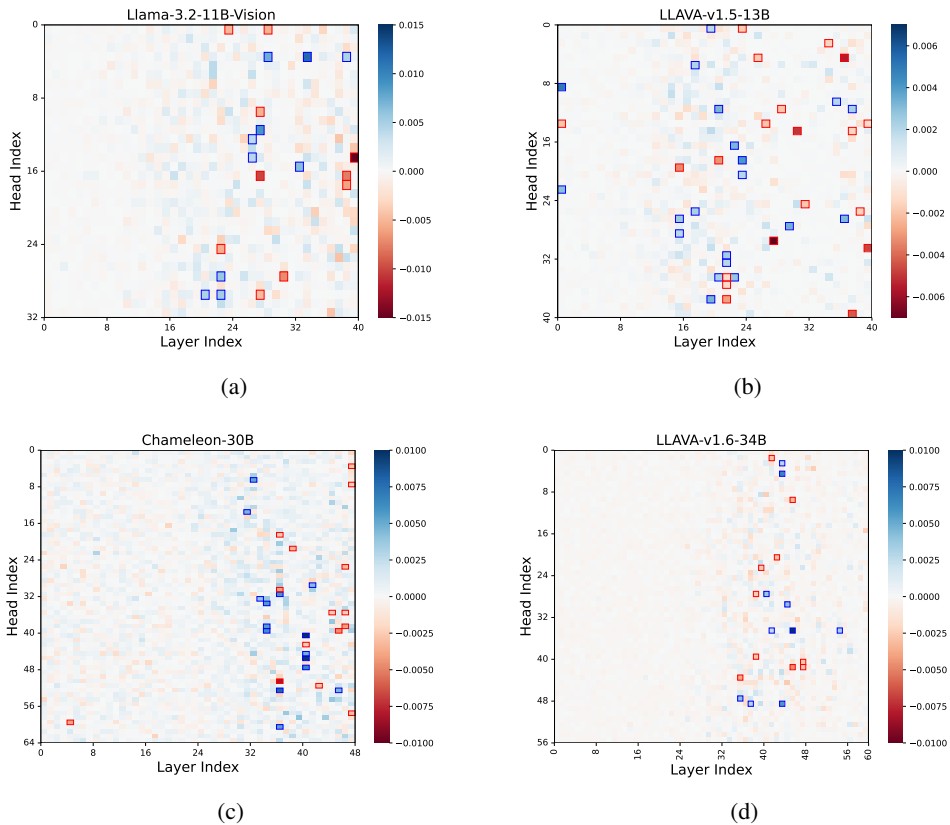

Figure 16: Contrastive influence maps of attention heads in Llama-3.2-11B-Vision, LLAVA-v1.5-13B, Chameleon-30B, and LLaVA-v1.6-34B. Bluer boxes indicate heads more responsible for generating hallucinated objects, and redder boxes indicate those more responsible for generating non-hallucinated objects.

Table 12: OPERA Hyperparameters.

| Parameters | Value |
|---|---|
| Self-attention Weight Scale Factor | 50 |
| Attending Retrospection Threshold | 15 |
| Beam Size | 5 |
| Penalty Weights | 1.0 |

Table 13: VCD Hyperparameters.

| Parameters | Value |
|---|---|
| Amplification Factor | 1 |
| Adaptive Plausibility Threshold | 0.1 |
| Diffusion Noise Step | 500 |

the official configurations provided by their respective sources. For the LURE baseline, we use the MiniGPT-4 13B checkpoint provided in the official repository as the revisor. We observed that, after generating the revised response, LURE's final step involves splitting the response into two parts based on the "\n" symbol and only retaining the first part. However, as most responses generated by LLaVA-7B contain "\n" in the middle, this split function significantly shortens the response length, reducing it by nearly 40% when the maximum generated token length is 128. This could compromise the fairness of the comparison. Therefore, we bypassed this post-processing step and directly used the full output response from the revisor as the final prediction for LURE.

Table 14: HALC Hyperparameters.

| Parameters | Value |
|---|---|
| Detector | Dino |
| JSD Buffer Size $m$ | 6 |
| Beam Size | 1 |
| Number of Sampled FOVs $n$ | 4 |
| Exponential Ratio of Growiing Contextual Field | 0.6 |
| Bounding Box Threshold for Dino | 0.4 |
| Adaptive Plausibility Threshold | 0.1 |

**Implementation details of Algorithm 1.** We select the top 20 attention heads as hallucination heads for LLaVA-7B and the top 10 heads for MiniGPT-4. For the threshold $\tau$ to control the when to deactivate text attention in decoding, we use a sweep search to find an optimal value. Based on this, we set $\tau = 0.4$ for LLaVA-7B and $\tau = 0.5$ for MiniGPT-4.

**Implementation details of Algorithm 2.** For LLaVA-7B model, we use the instruction-tuning dataset and fine-tuning codes from the offical Github repo[10]. The learning rate is set to $2 \times 10^{-5}$, and the global batch size is 128. We fine-tune the model for 200 steps, selecting the top 30 hallucination heads for fine-tuning. For the MiniGPT-4 model, we use the dataset and fine-tuning codes from the offical Github repo[11]. The learning rate is set to $3 \times 10^{-5}$, and the global batch size is 128. We fine-tune the model for 200 steps, selecting the top 20 hallucination heads for fine-tuning. For both models, the penalty weight $\lambda$ is set to 2. We only fine-tune the Query and Key matrices of the attention heads, as this operation modifies how values are linearly combined in self-attention without altering the basis of the linear space. We find that additionally fine-tuning the value matrices is ineffective (i.e., resulting in a higher hallucination rate of $\text{CHIRS}_\text{I}$ at 20.86% on LLaVA-7B and lower generation quality). We conjecture that this is because fine-tuning the value matrices significantly disrupts the linear space of the original model.

---

[10] https://github.com/haotian-liu/LLaVA?tab=readme-ov-file
[11] https://github.com/Vision-CAIR/MiniGPT-4

Table 15: Generated images description examples.

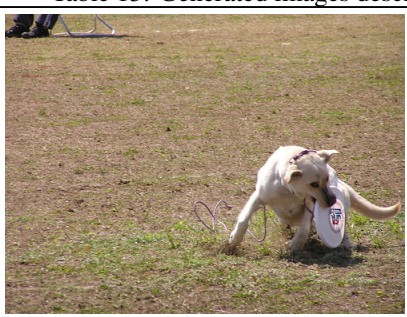

| Instruction | Please describe this image in detail. |
|---|---|
| Greedy | The image features a white dog playing with a frisbee in a grassy field. The dog is holding the frisbee in its mouth, running and enjoying the game. The frisbee is positioned towards the right side of the dog, and the dog appears to be in the middle of the field. \n\n In the background, there is a bench located on the left side of the field, possibly for spectators to sit and watch the dog play. |
| AD-HH (Ours) | The image features a white dog playing with a frisbee in a grassy field. The dog is holding the frisbee in its mouth, running and playing with the frisbee in the field. The dog appears to be enjoying the game and is actively engaged in the activity. |
| TF-HH (Ours) | The image features a white dog playing with a frisbee in a field. The dog is in the process of catching the frisbee, which is positioned in the middle of the scene. The dog is running towards the frisbee, displaying its agility and excitement. \n\n The field is spacious, providing ample room for the dog to play and enjoy its frisbee game. The dog's enthusiasm and the frisbee's position in the air create a lively and engaging scene. |

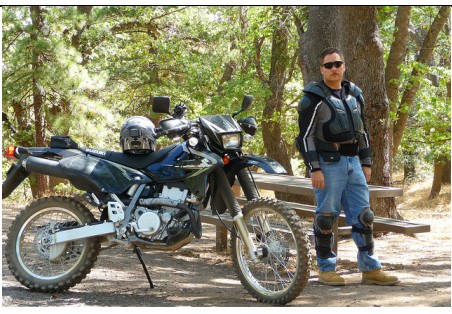

| Instruction | Please describe this image in detail. |
|---|---|
| Greedy | The image features a man standing next to a parked motorcycle, which is prominently displayed in the scene. The man is wearing a helmet and appears to be the owner of the motorcycle. The motorcycle is positioned in the foreground, occupying a significant portion of the image.\n\n In the background, there is a bench situated near the motorcycle, providing a place for the man to sit and relax. The scene also includes a backpack placed on the ground, possibly belonging to the man or someone else in the area. |
| AD-HH (Ours) | The image features a man standing next to a parked motorcycle, which is parked on a dirt road. The man is wearing a helmet and appears to be the owner of the motorcycle. The motorcycle is positioned on the left side of the image, with the man standing on the right side.\n\n In addition to the motorcycle, there is a bench located in the background, providing a place for the man to rest or take a break. The scene appears to be set in a wooded area, with trees surrounding the area. |
| TF-HH (Ours) | The image features a man standing next to a parked motorcycle, wearing a black jacket and sunglasses. He is posing for the camera, possibly in a wooded area. The motorcycle is parked on the side of the road, and the man is standing close to it.\n\n There are several trees in the background, and a bench can be seen further back in the scene. The man appears to be the main focus of the image, with the motorcycle and the surrounding environment serving as the backdrop. |

Table 16: Generated images description examples.

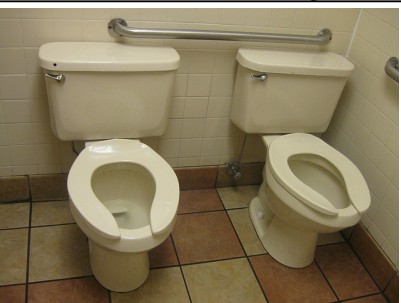

| Instruction | Please describe this image in detail. |
|---|---|
| Greedy | The image features a bathroom with two toilets placed side by side. Both toilets are white and have their lids up, revealing their bowls. The toilets are positioned close to each other, with one slightly closer to the left side of the room and the other slightly closer to the right side.\n\nThe bathroom also contains a sink, which is located towards the left side of the room. The sink is accompanied by a mirror, which is placed above it. The overall scene gives the impression of a public restroom or a shared bathroom space. |
| AD-HH (Ours) | The image features a bathroom with two toilets placed side by side in a bathroom stall. Both toilets are white and appear to be in good condition. The toilets are positioned close to each other, with one toilet slightly closer to the left side of the bathroom and the other toilet slightly closer to the right side.\n\nThe bathroom appears to be clean and well-maintained, with the toilets being the main focus of the image. The bathroom stall appears to be a public restroom, as it is designed to accommodate multiple users at. |
| TF-HH (Ours) | The image features a bathroom with two toilets placed side by side. Both toilets are white and appear to be in good condition. One of the toilets is positioned closer to the left side of the image, while the other is situated more towards the right side. \n\n The bathroom also features a tiled floor, which adds to the overall clean and well-maintained appearance of the space. |

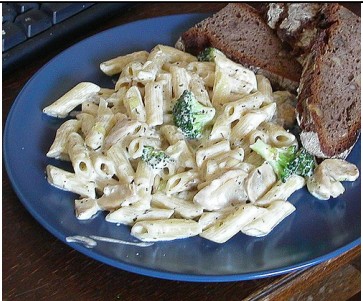

| Instruction | Please describe this image in detail. |
|---|---|
| Greedy | The image features a blue plate filled with a delicious meal consisting of pasta and broccoli. The pasta is mixed with a creamy sauce, and the broccoli pieces are scattered throughout the dish. The plate is placed on a dining table, and there is a sandwich on the table as well. \n\nIn addition to the main dish, there are two bowls in the scene, one located near the top left corner and the other near the top right corner. A keyboard can be seen in the background, suggesting that the dining area is part of a home office or work |
| AD-HH (Ours) | The image features a blue plate filled with a delicious meal consisting of pasta and broccoli. The plate is placed on a dining table, and the plate is filled with a generous portion of pasta and broccoli. The pasta appears to be a mix of macaroni and shells, and the broccoli is scattered throughout the dish.\n\nIn addition to the main dish, there is a slice of bread on the table, likely to be used as a side or to accompany the meal. The dining table occupies most of the image, and the dining |
| TF-HH (Ours) | The image features a blue plate filled with a delicious meal consisting of pasta and broccoli. The plate is placed on a wooden table, and the dish is accompanied by a piece of bread. The pasta and broccoli are arranged in a visually appealing manner, creating a colorful and appetizing presentation. The dish appears to be a healthy and nutritious meal, perfect for a meal or a light lunch. |

