# OpenReview forum: "Understanding and Mitigating Hallucination in Large Vision-Language Models via Modular Attribution and Intervention"
_ICLR.cc/2025/Conference — ICLR 2025 Poster_

### Official Review · Reviewer_xVEm · 2024-10-18

**Soundness:** 3
**Presentation:** 3
**Contribution:** 3
**Rating:** 8
**Confidence:** 5

**Summary:**

This study explores the causes of hallucination in large visual language models (LVLMs) during complex visual tasks and proposes mitigation measures. First, a causal analysis of specific modules of LVLM based on counterfactual editing found that the multi-head attention (MHA) module contributed more to the generation of hallucinatory words than the multi-layer perceptrons module. The study further identified attentional heads associated with hallucination, which are concentrated in the middle and deep layers of the model and show a strong attentional bias towards text markers. In addition, the patterns of these attentional heads remain relatively similar to the underlying language model and change more slowly during instruction tuning.

**Strengths:**

(1) It is the first work to analyze the attention map of the output text token of LVLMs and the attention-map of the NLP model, which is enlightening.

(2) This paper proposes two methods to alleviate the hallucination of multi-modal large model. One is to restrict the probability of generating tokens by improving transformer's decoder, and the other is to reduce over-reliance on text tokens by SFT. And the experiment proves that the two methods proposed in this paper are excellent.

**Weaknesses:**

(1) The training-free method is plug and play, but the methods applied in this paper are too few, only LLaVA and Minigpt4 are included, and the results of other models should be supplemented.

(2) The paper discusses the attention-map relationship of text tokens between LVLMs/LLM. Although the setting is a hallucination problem, this mechanism should be attributed to the cause of LLM. So can datasets be extended to other datasets like ScienceQA, GQA,textVQA, POPE, mmbench, etc.?

**Questions:**

(1) If it is a task like VQA, the generated text token is only one, and only ABCD is answered, whether this hallucination pattern does not exist. If it is the SFT method, then this result should also be applicable to VQA tasks?

(2) It is written in the paper that the author has hallucinatory attention-head in the middle layer or deep layer. Why is this? The complete distribution of 32 heads was not seen in the supplementary materials.

(3) Has it analyzed the difference between system_token+image token+prompt+output_token hallucination attention head and regular attention head? Is there any difference? Or can it only be observed through output-tokens?

(4)What about the distribution of attention when judging LLMs hallucinations? LLMs unable to generate a caption to an image, right? Therefore, I don't think this kind of attention head observation at the decoder terminal is very reasonable.

---

> ### Author Response · Authors · 2024-11-23
> **Response (Part I)**
>
> We sincerely thank you for taking the time to review our paper and for offering constructive feedback. We truly value your effort and have carefully addressed your concerns and questions below.
>
> **Comment 1**: The training-free method is plug and play, but the methods applied in this paper are too few, only LLaVA and Minigpt4 are included, and the results of other models should be supplemented.
>
> **Response 1**:  Thank you for your valuable feedback. To address your concerns, we have extended our experiments to include additional models, such as LLaVA-13B and Llama-3.2-11B-Vision, which are both modern and representative, with Llama-3.2-11B-Vision released just two months ago. Using the same settings on the COCO dataset, our method demonstrated significant improvements, reducing hallucination rates by approximately 6 points for Llama-3.2-11B-Vision and achieving a 10-point improvement in CHAIR_S for LLaVA-v1.5-13B. Here are the detailed results:
>
> | Method        | Llama3.2-11B-Vision |       | LLaVA-v1.5-13B |       |
> |---------------|----------------------|-------|----------------|-------|
> |               | CHAIR_S             | CHAIR_I | CHAIR_S       | CHAIR_I |
> | Greedy        | 28.4                | 7.4   | 48.6          | 12.4   |
> | AD-HH (Ours)  | **22.6**            | **4.9** | **38.8**      | **9.4** |
>
> We are actively working on expanding our experiments to include additional models and plan to incorporate these results in a future revision of the paper.
>
> **Comment 2**: The paper discusses the attention-map relationship of text tokens between LVLMs/LLM. Although the setting is a hallucination problem, this mechanism should be attributed to the cause of LLM. So can datasets be extended to other datasets like ScienceQA, GQA, textVQA, POPE, mmbench, etc.?
>
> **Response 2**: Thank you for your insightful question. The datasets you mentioned primarily involve short-form tasks with a focus on comprehension and discriminative features. For instance, POPE includes questions such as determining whether a chair or car exists in an image with a "yes" or "no" answer, where the object in question is explicitly mentioned in the prompt. These tasks differ fundamentally from open-ended generation tasks, where models must sequentially generate a series of tokens to complete the task without such explicit guidance in the prompt.  We applied our method, AD-HH, to these discriminative tasks and observed some improvement, though the gains were relatively modest:
>
> | Method        | ScienceQA | GQA    | TextVQA | POPE   |
> |---------------|-----------|--------|---------|--------|
> | Greedy        | 70.15     | 79.18  | 58.22   | 86.88  |
> | Ours  | **70.34** | **79.39** | 58.22   | **87.27** |
>
> The limited improvement may be attributed to the inherent differences between discriminative and generation tasks. In open-ended generation tasks, language bias becomes increasingly pronounced during the later stages of token generation [1][2], as the generated tokens deviate further from the input image tokens, leading to hallucinations. In contrast, in discriminative tasks, the generated tokens' position is closer to the image tokens and are less affected by such biases. This suggests that interventions targeting attention heads, like AD-HH, may not yield significant benefits for these tasks.
>
> [1] Yiyang Zhou et al. Analyzing and mitigating object hallucination in large vision-language models. ICLR, 2024
>
> [2] Qidong Huang et al. Opera: Alleviating hallucination in multi-modal large language models via over-trust penalty and retrospection-allocation. CVPR, 2024.
>
> **Question 3**: If it is a task like VQA, the generated text token is only one, and only ABCD is answered, whether this hallucination pattern does not exist. If it is the SFT method, then this result should also be applicable to VQA tasks?
>
> **Response 3**: Thank you for your question. We have included experimental results and a discussion on VQA tasks in Response 2.
>
> **Question 4**: It is written in the paper that the author has hallucinatory attention-head in the middle layer or deep layer. Why is this? The complete distribution of 32 heads was not seen in the supplementary materials.
>
> **Response 4**:  Thank you for your question. We are happy to clarify. Our conclusion that hallucination heads predominantly occur in the middle or deep layers is based on the complete distribution of all attention heads, as shown in Figure 2 (LLaVA-7B) and Figure 10 (MiniGPT-7B). These figures depict the contributions of all 1024 attention heads (from 32 layers with 32 heads per layer). The color coding in these figures represents each attention head's contribution to hallucination behavior. From the color distribution, it is evident that hallucination heads are concentrated in the middle and deep layers.

---

> ### Author Response · Authors · 2024-11-23
> **Response (Part II)**
>
> **Question 5**: Has it analyzed the difference between the attention patterns of hallucination heads and regular attention heads across system tokens, image tokens, prompt tokens, and output tokens? Is there any distinction, or can differences only be observed through output tokens?
>
> **Response 5**: Thank you for your question. We believe you are referring to Figure 4, which presents the attention patterns of hallucination and non-hallucination heads, albeit limited to output tokens. To address your concern, we have included an additional analysis in Figure 13 in the Appendix, which examines attention patterns across the entire sequence, encompassing system tokens, image tokens, prompt tokens, and output tokens. Please let us know if we have misunderstood your question.
>
> Specifically, Figure 13 in the Appendix highlights that hallucination heads allocate significantly more attention weight to text tokens, often neglecting visual tokens. In contrast, non-hallucination heads distribute attention more evenly, with a greater focus on visual tokens. These findings are consistent with the observations in Figure 4 and further support the conclusion that hallucination and non-hallucination heads exhibit distinct attention patterns when processing input sequences.
>
> **Question 6**: What about the distribution of attention when judging LLMs hallucinations? LLMs unable to generate a caption to an image, right? Therefore, I don't think this kind of attention head observation at the decoder terminal is very reasonable.
>
> **Response 6**:  Thank you for your question. If we understand correctly, you are drawing a connection between the hallucination heads identified in LVLMs and their behaviors in LLMs, and asking whether our conclusions extend to LLMs. Please let us know if we’ve misunderstood your intent.
>
> To clarify, hallucinations in LVLMs are fundamentally different from those in LLMs, and our conclusions do not directly apply to hallucinations in LLMs. Specifically, LVLMs generate responses by integrating visual and textual information. The hallucinations we observed are caused by an imbalance in attention between image and text tokens in certain hallucination heads. This imbalance likely arises because LVLMs are fine-tuned from LLMs, where language bias tends to dominate during question answering, causing the model to undervalue visual input. Thus, this issue is inherently tied to modality competition between vision and language in LVLMs.
>
> In contrast, hallucinations in LLMs occur in a single-modal setting and are primarily related to factors within in the language model itself, which requires separate investigation.  Our findings are specific to multimodal models and do not generalize to LLMs in isolation.
>
> ***
> We sincerely thank you for your thoughtful review. We hope the revised manuscript and the clarifications provided above have addressed your concerns effectively. If our responses have satisfactorily resolved your concerns, we would greatly appreciate it if you could consider updating your review score. However, if you have any remaining concerns or require further clarification, please do not hesitate to let us know. We are more than willing to provide additional explanations or updates as needed.

---

> > ### Comment · Reviewer_xVEm · 2024-11-23
> > **Some question remain unsolved**
> >
> > Thanks for the author's reply, but I still have some questions that have not been resolved yet.Q1.Figure 13 defines the left image as an hallucination head, is this scientific?Q2.After visualizing LLaVA through the attention map, it will be found that there are some  attention distributions of the mid/deep layers that are similar to the left one, which puzzles me to know how to define these Q3.Are sparsely and unevenly distributed heads unhelpful for inference? Although this is hard to explain for black-box models.Q4.Can it be shown that TF-HH works for different projector models like Instruct-Blip2 and Qwen-VL?
> > I'd be happy to raise points if the authors can address these queries of mine above.

---

> > > ### Author Response · Authors · 2024-11-26
> > > **Response for Reviewer xVEm (Part I)**
> > >
> > > Thank you for sharing your valuable feedback! We’re glad to address your concerns.
> > >
> > > **Question 1**: Figure 13 defines the left image as an hallucination head, is this scientific?
> > >
> > > **Response 1**:  We believe there may be a potential misunderstanding, and we would like to clarify. You might assume that we define attention heads with attention maps similar to the left part of Figure 13 as hallucination heads. However, this is not the case. In our paper, we define hallucination heads as those most responsible for hallucination behaviors, identified through their contrastive influence scores, as described in Equation (2). This definition is based on counterfactual analysis [1], which we believe is scientifically sound.
> > >
> > > Subsequently, we analyze the behaviors of these identified hallucination heads and observe their significant over-reliance on text tokens in attention maps (e.g., the left part of Figure 13). It is important to emphasize that we do not arbitrarily classify heads with attention maps resembling the left part of Figure 13 as hallucination heads.
> > >
> > > [1] J. Pearl. Causality: Models, Reasoning, and Inference. 2000
> > >
> > > **Question 2**: After visualizing LLaVA through the attention map, I noticed that some attention distributions in the mid/deep layers resemble the one on the left, which leaves me puzzled about how to define these.
> > >
> > > **Response 2**: We understand your are worring about whether a particular attention head, which displays similarities to the left attention map in Figure 13 (e.g., over-reliance on text tokens), can be classified as a hallucination head. Our answer is no. Let us clarify:
> > > - In our paper, hallucination heads are identified based on their causal effects on hallucination behavior, specifically through large contrastive influence values. By definition, these heads directly contribute to hallucinations.
> > > - Empirically, we identified attention heads with largest contrastive influence values, especially the top 20 ones in LLaVA-7B to analyze their behaviors. We find that these hallucination heads consistently exhibit over-reliance on text token inputs. However, we emphasize that this over-reliance is a symptom of hallucination heads, not a criterion to identify them.
> > > - We note that there may be other attention heads that also exhibit over-reliance on text tokens but do not influence hallucination behavior (e.g., layer 31, head 4 in LLaVA-7B). This occurs because some heads may function as a general-purpose language head, ensuring fluency and coherence in text generation. Although these heads look like the left attention map in Figure 13 and over-rely on text tokens, they are unrelated to hallucination behavior.
> > > - To support the above claims, we have conducted a statistical analysis of all 1024 attention heads in LLaVA-7B, 412 attention heads (~40%) exhibit heavy reliance on text tokens  (text attention/image attention>3). Of these 412 heads, 18 are identified as hallucination heads based on their causal contribution to hallucination behavior. In total, there are 20 hallucination heads, meaning most (90%) exhibit over-reliance on text tokens.
> > >
> > > In summary, hallucination heads often show over-reliance on text tokens. However, not all heads that over-rely on text tokens are hallucination heads. They qualify as hallucination heads only when they are causally responsible for hallucination behavior, indicated by large contrastive influence values. Please let us know whether the response above can address your concerns and we are more than willing to clarify.
> > >
> > > **Question 3**: Are sparsely and unevenly distributed heads unhelpful for inference? Although this is hard to explain for black-box models.
> > >
> > > **Response 3**: We understand your concern about whether a particular attention head, which appears sparse and unevenly distributed (similar to the left attention map in Figure 13), is unhelpful for inference. As discussed in Response 2, the answer is no: there is no direct correlation or one-to-one mapping between sparse, uneven attention patterns and a head’s contribution to response generation.
> > >
> > > While hallucination heads often exhibit sparse and uneven attention distributions, but it does not lead to the conclusion that all heads with sparse and uneven distributions are unhelpful for inference. Sparse and uneven patterns may occur naturally in attention heads that contribute to general language generation purposes, such as ensuring coherence, fluency in generation.
> > > Therefore, sparse and uneven attention distribution alone cannot be used as a single criterion to classify a head as unhelpful for inference.

---

> > > ### Author Response · Authors · 2024-11-26
> > > **Response for Reviewer xVEm (Part II)**
> > >
> > > **Question 4**: Can it be shown that TF-HH works for different projector models like Instruct-Blip2 and Qwen-VL? I'd be happy to raise points if the authors can address these queries of mine above.
> > >
> > > **Response 4**: Sure. We have successfully applied our method, TF-HH, to the Qwen-VL model. By specifically fine-tuning hallucination heads, we successfully reduced the hallucination rate by 4.4% on CHAIR_S and 2.2% on CHAIR_I, while maintaining a comparable BLEU score. This result demonstrates the effectiveness and general applicability of TF-HH to other models.
> > > | Qwen-VL         | CHAIR_S | CHAIR_I | BLEU |
> > > |------------------|---------|---------|------|
> > > | Greedy          | 38.6    | 12.1    | 19.5 |
> > > | TF-HH (Ours)    | **34.2** | **9.9** | **20.3** |
> > >
> > > Details: We use 1000 samples from COCO to identify hallucination heads and apply TF-HH to fine-tune the top 20 hallucination heads. Our training setup is similar to that used for fine-tuning LLaVA-7B, with the exception that we do not have access to Qwen-VL's training data. Therefore, we use LLaVA’s training data and convert it into the format required for Qwen-VL. We apply the same learning rate and train for 200 iterations.
> > >
> > > ***
> > > We hope our responses have sufficiently addressed your questions, and we have updated our manuscript to clarify sections that may have caused confusion. Thank you for your valuable feedback!

---

> > > > ### Comment · Reviewer_xVEm · 2024-11-26
> > > > **Response for author**
> > > >
> > > > The author solved my question, the discussion was very beneficial, and I will raise my score.

---

> > > > > ### Author Response · Authors · 2024-11-26
> > > > > **Thanks for your positive feedback**
> > > > >
> > > > > Thank you so much for the positive feedback, we are more than appreciated!  Your valuable participation and suggestions have contributed to improving the quality of our paper. Thank you once again for your time and effort!

---

### Official Review · Reviewer_Ex6k · 2024-11-04

**Soundness:** 4
**Presentation:** 3
**Contribution:** 4
**Rating:** 8
**Confidence:** 5

**Summary:**

The paper addresses the issue of hallucination in Large Vision-Language Models (LVLMs), specifically focusing on why these models generate content that deviates from the provided image information in open-ended tasks like image captioning. The authors conduct a systematic investigation using causal mediation analysis and counterfactual edits to identify the internal components responsible for hallucination. They find that Multi-Head Attention (MHA) modules contribute more to hallucination than Multi-Layer Perceptron (MLP) modules, and within MHAs, certain attention heads—termed "hallucination heads"—are primarily responsible. To mitigate hallucination, the paper proposes two methods: (1) an adaptive deactivation of hallucination heads during decoding, which is training-free and can be applied directly, and (2) targeted fine-tuning of hallucination heads to reduce their reliance on text tokens. Both methods demonstrate significant reductions in hallucination rates on benchmark datasets like COCO captioning and Nocaps, outperforming existing baselines.

**Strengths:**

- The paper deploys a systematic step-by-step approach to identify the components responsible for hallucinations. The insights are valuable and the analysis seems sensible.
- Identifying that only specific attention heads seem to contribute to hallucinations is a novel finding. The additional analysis they did (fig 5) is really interesting.
- The mitigation strategies shown (both training-based and training-free) seem sensible and seem to work well on the benchmarks.
- They show the method working with multiple datasets and multiple models.

**Weaknesses:**

- There is limited discussion on how the proposed interventions affect the model's overall language generation capabilities. Potential trade-offs between reducing hallucination and maintaining fluency or coherence are not thoroughly examined.
- The paper is almost entirely focused on object hallucination.
- The experiments are conducted on 7B parameter models (LLaVA-7B and MiniGPT-4). Given the trend towards larger models in the field, it would be valuable to assess whether the identified hallucination heads and mitigation strategies are applicable to larger models (e.g., 70B parameters) and whether similar patterns emerge at different scales.
- There are minor issues with the writing and presentation that could be improved for clarity and professionalism. For example, phrases like "Our Method Run Fast in Generation" could be rephrased for better readability.

**Questions:**

- Can you provide more details on why certain hallucination heads exhibit slow changes during instruction tuning? What factors contribute to this "laziness," and how might future work address this issue?
- Have you evaluated how the proposed interventions affect the overall language generation quality of the models? Specifically, does reducing reliance on text tokens in hallucination heads impact the fluency, coherence, or descriptiveness of the generated captions? It would be helpful to see metrics (human eval) or analyses addressing potential trade-offs.
- Do you think "hallucination heads" exist in the same way in larger scale (eg. 70b) VLMs? Would the same training-free method work similarly on them? Would be interesting to see.

---

> ### Author Response · Authors · 2024-11-23
> **Response (Part I)**
>
> Thank you for reading our paper and providing valuable feedback. We greatly appreciate your comments and have addressed your concerns and questions below.
>
> **Comment 1**: There is limited discussion on how the proposed interventions affect the model's overall language generation capabilities. Potential trade-offs between reducing hallucination and maintaining fluency or coherence are not thoroughly examined.
>
> **Response 1**:  Thank you for raising this concern. We acknowledge the potential trade-offs between reducing hallucination and maintaining the overall generation quality. This trade-off was observed when applying the initial naive method in our experiments, as discussed in Section 4.3.1 (refer to Figure 6).
>
> To address this issue, we developed approaches that achieve a more balanced trade-off, such as adaptive deactivation of hallucination heads and targeted fine-tuning. Supplementary results in Table 4, Figures 14 and 15 provide a detailed analysis of these trade-offs, illustrating how varying hyperparameters in our methods impacts both hallucination reduction and generation quality.
>
> **Comment 2**: The paper is almost entirely focused on object hallucination.
>
> **Response 2**: Thank you for your comment. Object hallucination is a significant challenge for LVLMs in open-ended generation tasks, which is why we chose to focus on it in this study. Nevertheless, our method is not limited to object hallucination and can be extended to address other types of hallucination, such as counting and positional errors. For instance, on the MME benchmark, we observed a 5-point improvement in counting performance (from 148 to 153), a 10-point improvement in positional accuracy (from 128 to 138). Detailed results are reported below:
>
> |              | existence | count   | position | color | posters | celebrity | scene   | landmark | artwork | OCR     |
> | ------------ | --------- | ------- | -------- | ----- | ------- | --------- | ------- | -------- | ------- | ------- |
> | Baseline     | 190       | 148     | 128      | 160   | 139     | 133       | 156     | 162      | 122     | 130     |
> | AD-HH (Ours) | 190       | **153** | **138**  | 160   | **142** | **135**   | **158** | 162      | 118     | **138** |
>
> **Comment 3**: The experiments are conducted on 7B parameter models (LLaVA-7B and MiniGPT-4). Given the trend towards larger models in the field, it would be valuable to assess whether the identified hallucination heads and mitigation strategies are applicable to larger models (e.g., 70B parameters) and whether similar patterns emerge at different scales.
>
> **Response 3**: Thank you for this insightful comment. Due to limited computational resources, our initial experiments focused on 7B parameter models. We completely agree that investigating hallucination behaviors in larger models and studying scaling and emergence effects is interesting and important. To address your concern, we extended our experiments to larger models, including Llama-3.2-11B-Vision and LLaVA-v1.5-13B. Unfortunately, resource and time constraints prevented us from exploring 70B parameter models. Nevertheless, our findings indicate that the proposed method is effective on 13B-sized models. Comparative results for hallucination on the COCO dataset are provided below.
>
> | Method        | Llama3.2-11B-Vision |       | LLaVA-v1.5-13B |       |
> |---------------|----------------------|-------|----------------|-------|
> |               | CHAIR_S             | CHAIR_I | CHAIR_S       | CHAIR_I |
> | Greedy        | 28.4                | 7.4   | 48.6          | 12.4   |
> | AD-HH (Ours)  | **22.6**            | **4.9** | **38.8**      | **9.4** |
>
> **Comment 4**: There are minor issues with the writing and presentation that could be improved for clarity and professionalism. For example, phrases like "Our Method Run Fast in Generation" could be rephrased for better readability.
>
> **Response 4**: Thank you for your suggestion. We have revised the paper to enhance clarity.

---

> ### Author Response · Authors · 2024-11-23
> **Response (Part II)**
>
> **Question 5**: Can you provide more details on why certain hallucination heads exhibit slow changes during instruction tuning? What factors contribute to this "laziness," and how might future work address this issue?
>
> **Response 5**:  Thank you for the thoughtful questions. We address them separately below.
>
> 1. Why do certain hallucination heads exhibit slow changes during instruction tuning?
>
> Our analysis of gradient norms for these hallucination heads revealed that their gradients are consistently smaller compared to non-hallucination heads throughout training. For instance, at iteration 0, the gradient norm for hallucination heads was 1.3 times smaller than that of non-hallucination heads. This trend persisted across multiple iterations (e.g., 500, 1000, 2000, 3000, 4000, and 5000), explaining their slower updates during training.
>
> We hypothesize that this "laziness" arises from over-optimization during pre-training and instruction tuning of the base language model. Over time, certain attention heads may become less responsive to new inputs, such as multi-modal data in LVLMs. Ideally, landscape analysis could provide insight into the sharpness and plasticity of these attention heads. However, existing visualization techniques are not yet scalable to 7B models, limiting this approach.
>
> 2. How might future work address this issue?
>
> Addressing this challenge requires careful fine-tuning, particularly in low-data scenarios where models tend to inherit shortcut patterns from their base language models. These inherited biases can amplify hallucinations in open-ended tasks, making it crucial to adapt LVLMs to specific tasks and datasets effectively.
>
> We highlight a few future directions:
> * From the perspective of training algorithms, future research could explore targeted mitigation strategies inspired by continual learning literature (see e.g., [1]).
> * From the representation perspective, techniques such as model expansion (see e.g., [2,3]), which introduce more flexible components, could enhance adaptability.
> * From the data perspective, additional efforts to curate and augment datasets could further improve performance and robustness.
>
> [1] Dohare, Shibhansh, et al. "Loss of plasticity in deep continual learning." Nature 632.8026 (2024): 768-774.
>
> [2] Yoon, Jaehong, et al. "Lifelong learning with dynamically expandable networks." ICLR 2018.
>
> [3] Anagnostidis, Sotiris, et al. "Navigating Scaling Laws: Compute Optimality in Adaptive Model Training." ICML 2024.
>
> **Question 6**: Have you evaluated how the proposed interventions affect the overall language generation quality of the models? Specifically, does reducing reliance on text tokens in hallucination heads impact the fluency, coherence, or descriptiveness of the generated captions? It would be helpful to see metrics (human eval) or analyses addressing potential trade-offs.
>
> **Response 6**:  Thank you for the suggestion. To evaluate the impact of our interventions, we conducted a manual assessment involving two Ph.D. students and one undergraduate student. They evaluated the responses using a 1 to 5 scoring system based on two criteria: (1) Non-hallucination performance, where higher scores reflect fewer hallucinations, and (2) Generation quality, where higher scores indicate more fluent and coherent outputs.
>
> | Method           | Non-Hallucination Score | Generation Quality Score |
> |-------------------|-------------------------|---------------------------|
> | Greedy Decoding  | 3.25                    | 3.99                      |
> | AD-HH (Ours)     | **3.87**                    | 3.85                      |
> | FT-HH (Ours)     | 3.78                    | **4.01**                      |
>
> For both the baseline and our proposed methods, the evaluators assessed a total of 500 generated responses per method, resulting in 1500 responses overall. These results demonstrate that our methods effectively mitigate hallucination while maintaining high generation quality.

---

> > ### Comment · Reviewer_Ex6k · 2024-11-23
> >
> > Also, in the table above on OpenReview you refer to the method as FT-HH while in the paper it is TF-HH. I assume those are the same thing and I wrote my comment with that assumption.

---

> ### Author Response · Authors · 2024-11-23
> **Response (Part III)**
>
> **Questions 7**: Do you think "hallucination heads" exist in the same way in larger scale (eg. 70b) VLMs? Would the same training-free method work similarly on them? Would be interesting to see.
>
> **Response 7**: Thank you for raising this thought-provoking question. We have extended our method on larger models including Llama-3.2-11B-Vision and LLaVA-v1.5-13B, which still works on them. It is indeed exciting to consider whether hallucination heads behave differently or even vanish in larger-scale models like 70B. Unfortunately, we currently lack the computational resources and time to study this topic. However, we can share some preliminary thoughts that future work may explore further.
>
> First, we note that large models differ from small models in both representation power and optimization dynamics. Intuitively, larger models are expected to possess greater representation capacity. Additionally, neural network learning theory, such as the Neural Tangent Kernel framework, suggests that larger models tend to change their parameters slowly at each training step. Building on this foundation, we propose two potential scenarios based on the extent of training:
>
> - Short-training regime: In this scenario, larger models may strongly inherit pre-existing language biases, and hallucination heads are likely to exist. This outcome is expected when the model is fine-tuned on small-scale dataset to adapt to specific downstream tasks, especially if the language model backbone has been extensively trained on text data alone.
> - Long-training regime: With sufficient training, larger models can utilize their superior representation power to learn accurate patterns from extensive datasets—something smaller models often struggle to achieve. Hallucination heads are expected to disappear in this scenario.
>
> In conclusion, we cannot provide a definitive answer at this time. The persistence or mitigation of hallucination heads likely depends on the scale of training efforts. This relationship is challenging to predict in advance but is an interesting area for future research. We hope our study can provide some insights and plan to study this topic in the future.
>
> ---
>
> We sincerely thank you for your valuable review comments. We hope the revised manuscript and the clarifications provided have effectively addressed your concerns. If our responses meet your expectations, we would greatly appreciate your consideration in updating your review score. Should you have any remaining questions or require further clarification, please do not hesitate to reach out. We are more than happy to provide additional explanations or updates as needed.

---

> ### Comment · Reviewer_Ex6k · 2024-11-23
>
> Thank you very much for the additional experiments and commentary. I feel more confident now about the results presented in the paper. I feel like the rating for this paper now may be updated to 7 which is unfortunately not an option here.
>
> 1- However, I still believe that showing the scalability of this method (at least to the 30B~) scale would be very substantial. I understand that it may require a lot of VRAM to analyze a 70B, but it would be interesting to analyze a more intermediate model like LLaVa[1] 34B
>
> 2- Have you also explored "early fusion" approaches such as Chameleon[2] (https://huggingface.co/facebook/chameleon-7b https://huggingface.co/facebook/chameleon-30b) and whether hallucination heads exist there? A negative result would be fine and could make sense.
>
> 3- Are you willing, upon paper acceptance, to release open-source code for AD-HH, FT-HH, as well as for the analysis shown in Sec. 4 of your paper so that future works may further investigate the very interesting phenomenon you present in this paper? Open source weights for FT-HH models would also be interesting.
>
> If any one of those points are addressed I would be more than happy to update my score.
>
> [1] Liu, Haotian, et al. "Visual instruction tuning." Advances in neural information processing systems 36 (2024).
> [2] Team, Chameleon. "Chameleon: Mixed-modal early-fusion foundation models." arXiv preprint arXiv:2405.09818 (2024).

---

> ### Author Response · Authors · 2024-11-25
> **Response for Reviewer Ex6k (Part I)**
>
> Thank you for your feedback. We are pleased to hear that our previous responses addressed your concerns. We apologize for the typo of "FT-HH" in the previous response, and we have corrected it. Below, we have provided responses to your new questions:
>
> **Question 1**: However, I still believe that showing the scalability of this method (at least to the 30B~) scale would be very substantial. I understand that it may require a lot of VRAM to analyze a 70B, but it would be interesting to analyze a more intermediate model like LLaVa[1] 34B.
>
> **Response 1**:  We understand your concern and have tried our best to study the LLaVA-v1.6-34B (LLaVA-34B for short) model. We have applied our method to attribute and intervention on this model and found reduced hallucination rate. Detailed results are presented below:
>
> First, we evaluate the hallucination performance of LLaVA-34B on the COCO dataset. For LLaVA-34B, the hallucination rate is 23.2% for CHAIR_S and 6.4% for CHAIR_I,  which are smaller than that of  LLaVA-v1.5-7B (LLaVA-7B for short) and LLaVA-v1.5-13B (LLaVA-13B for short). According to the technical report of LLaVA-34B, these improvements can be attributed to a better LLM backbone and an increased input image resolution. We observe a general trend of decreasing hallucination rates as model size increases. However, as noted, this improvement is likely to be influenced by multiple factors.
>
> |                            | LLaVA-7B | LLaVA-13B | LLaVA-34B |
> |----------------------------|----------|-----------|-----------|
> | Hallucination Rate (CHAIR_S) | 51.8     | 48.6      | 23.2      |
> | Hallucination Rate (CHAIR_I) | 13.3     | 12.4      | 6.4       |
>
> Second, we apply our modular attribution to analyze hallucination heads across different scales of LVLMs. To fairly compare the number of hallucination heads across models of varying scales, we identify salient hallucination heads —those with contrastive influence values exceeding 25% of the maximum contrastive influence value for each model. The 25% threshold is temporarily selected for comparison purpose. We evaluate both their absolute numbers and their ratio relative to all attention heads.
>
> |                               | LLaVA-7B | LLaVA-13B | LLaVA-34B |
> |-------------------------------|----------|-----------|-----------|
> | Total Number of Attention Heads | 1024     | 1600      | 3360      |
> | Number of Salient Hallucination Heads | 42       | 37        | 10        |
> | Ratio of Salient Hallucination Heads  | 4.1%     | 2.3%      | 0.3%      |
>
> Our findings above reveal that hallucination heads tend to diminish as model size increases and sufficient post-training is applied. Although we cannot perfectly isolate the contributions of individual factors (e.g., the LLM backbone, data size and sources, image tokenizer), our observations tend to align with our hypothesis: larger models possess stronger representational power to learn correct behaviors from data, whereas smaller models are more susceptible to language bias.
>
> Finally, we apply our targeted modular intervention method  to LLaVA-34B and find that the hallucination rate can be reduced by 2.8% on CHAIR_S and 0.8% for CHAIR_I. Note that this improvement is not easy given the superior performance of LLaVA-34B. We also perform preliminary generation quality evaluation and found  that the BLEU score is comparable with the one before intervetion.
>
> | LLaVA-34B | CHAIR_S | CHAIR_I | BLEU  |
> |----------------|---------|---------|-------|
> | Greedy         | 23.2    | 6.4     | 15.1  |
> | AD-HH (Ours)           | **20.4** | **5.6** | **15.2** |
>
> Thank you for initiating the discussion on model behaviors, particularly regarding model size scaling. We have updated our paper to include these results in Appendix A.3. We believe these findings could help us better understand hallucinations in LVLMs and hope they could be valuable to the community.

---

> ### Author Response · Authors · 2024-11-25
> **Response for Reviewer Ex6k (Part II)**
>
> **Question 2**:  Have you also explored "early fusion" approaches such as Chameleon [2] (https://huggingface.co/facebook/chameleon-7b https://huggingface.co/facebook/chameleon-30b) and whether hallucination heads exist there? A negative result would be fine and could make sense.
>
> **Response 2**: Thank you for highlighting the "early fusion" model. We appreciate the work on Chameleon, which offers a comprehensive study of early fusion-based multi-modal training, enabling reasoning and generation across modalities using shared representations. Chameleon-30B is easy to use and achieve state-of-the-art performance on many tasks, but we have observed that it still exhibits hallucination behaviors.
>
> First, we observed that Chameleon-30B has a hallucination rate of 38.0% on CHAIR_S and 12.6% on CHAIR_I on the COCO dataset. Next, we identified the top 10 most salient hallucination heads in Chameleon-30B, as shown in Figure 16(c) in the Appendix. By deactivating these identified hallucination heads, we successfully reduced the hallucination rate on CHAIR_S from 38.0% to 34.8%, with a comparable BLEU score.
>
> | Chameleon-30B | CHAIR_S | CHAIR_I | BLEU  |
> |---------------|---------|---------|-------|
> | Greedy        | 38.0    | 12.6    | **10.9**  |
> | AD-HH (Ours)  | **34.8**    | **12.5**    | 10.8 |
>
> **Question 3**: Are you willing, upon paper acceptance, to release open-source code for AD-HH, FT-HH, as well as for the analysis shown in Sec. 4 of your paper so that future works may further investigate the very interesting phenomenon you present in this paper? Open source weights for FT-HH models would also be interesting.
>
> **Response 3**: Absolutely! Upon paper acceptance, we will release the code and model weights necessary to reproduce our analysis and experiment results. We hope this will benefit the research community and inspire further investigation.
>
> ***
> We sincerely thank you again for your valuable review comments. We hope our responses have adequately addressed your questions, and we are happy to provide further clarification if needed.

---

> > ### Comment · Reviewer_Ex6k · 2024-11-25
> >
> > Dear Authors,
> >
> > Thank you very much for the additional results. I realize that this must've been difficult to complete so quickly. I've updated my score.
> >
> > Some further notes:
> > -It would be interesting to also present the number and ratio to salient hallucination heads in Chameleon at multiple scales (8b, 30b..) to confirm that the pattern is similar to LLaVa.
> > -It would also be interesting to test all of these models on MME to verify how universal the results are and whether the same ratios/patterns seen in CHAIR would replicate with the different benchmark.
> >
> > Over all, I think this is a good paper with a very insightful analysis and I'm thankful to the authors for presenting it and for being so proactive throughout the review process.

---

> > > ### Author Response · Authors · 2024-11-25
> > > **Thanks for your positive feedback**
> > >
> > > Thank you for your positive feedback, which is greatly appreciated! We have put significant effort into studying 30B-size models, and the empirical results are exciting for us. We are currently working to address the concerns raised by other reviewers. We will explore the Chameleon-7B model and include a more detailed analysis in the future, as we recognize the importance of these results. Thank you once again for your understanding and support!

---

### Official Review · Reviewer_JmFc · 2024-11-04

**Soundness:** 3
**Presentation:** 3
**Contribution:** 2
**Rating:** 6
**Confidence:** 4

**Summary:**

This work empirically studies the hallucination problem of LVLMS via counterfactual analysis. This work reveals multiple interesting findings on why hallucination occurs and how to mitigate it. The analyses are performed with two models (LLaVA-7B and MiniGPT-4) on two benchmark datasets (COCO and MM-Vet).

**Strengths:**

1. This work shows several interesting observations about hallucination, as summarized in the sentences in the bold font in section 4.

2. Based on the findings, this work proposes two simple ideas to mitigate hallucination – adaptive deactivation and targeted fine-tuning of hallucination heads.

**Weaknesses:**

1. This work fails to cite several related recent works as follows, to name a few.
- They may need to be cited and compared in experiments.
- A. Deng et al., Seeing is Believing: Mitigating Hallucination in Large Vision-Language Models via CLIP-Guided Decoding, arXiv 2024.
- F. Liu et al., Mitigating Hallucination in Large Multi-Modal Models via Robust Instruction Tuning, ICLR 2024.
- J. Zhang et al., Reflective Instruction Tuning: Mitigating Hallucinations in Large Vision-Language Models, ECCV 2024.

2. The evaluation only depends on two CHAIR metrics.

3. Some findings look obvious.
- It is not so surprising that the multi-head-attention is more critical than feed-forward network, since the former takes a majority of parameters and does much more things than the latter in the transformer.
- It is a well-known fact that hallucination is more related to language bias (rather than) image bias and multimodal models often ignore the information from images compared to that from text.

**Questions:**

1. More in-depth analysis would be required to discuss why the fine-tuning approach is not as good as the training-free one.

---

> ### Author Response · Authors · 2024-11-23
> **Response (Part I)**
>
> Thank you for taking the time to review our paper and provide valuable feedback. We greatly appreciate your efforts and have addressed your concerns and questions below.
>
> **Comment 1**:  This work fails to cite several related recent works. They may need to be cited and compared in experiments.
>
> **Response 1**:  Thank you for bringing these works to our attention. We have added a discussion of these studies to the related work section and appreciate their contributions:
>
> - [Deng et al., 2024] proposed a  CLIP-guided decoding approach that utilizes CLIP as an external tool to alleviate hallucination during decoding.
> - [Liu et al., 2024] addressed the issue by constructing the Large-scale Robust Visual (LRV)-Instruction dataset, which includes both positive and negative instructions to enhance the robustness of visual instruction tuning.
> - [Zhang et al., 2024] introduced a large-scale instruction-tuning dataset name REVERIE with reflective rational annotations, to enable the model to justify whether the reponses are correct or incorrect.
>
> Different from these works, our work takes a different direction by specifically identifying, analyzing, and adapting the components within the model responsible for hallucinations. Additionally, we have conducted experiments to empirically compare our method with these baseline approaches on the LLaVA model (see Table 8 in the Appendix ). The results below demonstrate that, although these baselines also help reduce hallucination errors, our method, which employs targeted interventions, is more effective in mitigating object hallucinations in open-ended generation tasks.
>
> |           | Greedy | GCD [Deng et al., 2024] | LRV [Liu et al., 2024] | REVERIE [Zhang et al., 2024] | AD-HH (Ours) | TF-HH (Ours) |
> |-----------|--------|--------------------------|-------------------------|-----------------------------|--------------|--------------|
> | CHAIR_S   | 51.8   | 39.2                     | 39.4                    | 49.6                        | **29.6**         | 35.0         |
> | CHAIR_I   | 13.3   | 10.8                     | 13.1                    | 12.7                        | **8.0**         | 8.7          |
>
> Our current evaluation focuses on the LLaVA model, as the results of these methods are not directly applicable to our settings due to differences in model versions and evaluation protocols (details provided in the Appendix A). Consequently, re-implementing these baselines is necessary, and some methods, such as LRV and REVRIE, require extensive training that demands significant computational resources. We are actively conducting further assessments with additional methods and settings and plan to share updated findings later. Nevertheless, the current evidence strongly supports the effectiveness of our proposed methods in mitigating hallucinations.
>
>
> **Comment 2**: The evaluation only depends on two CHAIR metrics.
>
> **Response 2**:  Thank you for raising this concern. To address your concern, in addition to the two CHAIR metrics used to evaluate the effectiveness of our method in mitigating hallucination, we have also conducted a human evaluation as suggested by **Reviewer Ex6k**.
>
> | Method           | Non-Hallucination Score | Generation Quality Score |
> |-------------------|-------------------------|---------------------------|
> | Greedy Decoding  | 3.25                    | 3.99                      |
> | AD-HH (Ours)     | **3.87**                    | 3.85                      |
> | TF-HH (Ours)     | 3.78                    | **4.01**                      |
>
> Specifically, we asked two Ph.D. students and one undergraduate student to manually evaluate the responses. They were instructed to score each response on a scale of 1 to 5 based on two criteria: (1) non-hallucination performance, with higher scores reflecting fewer hallucinations, and (2) generation quality, with higher scores indicating more fluent responses. For both the baseline and our proposed methods, the evaluators assessed a total of 500 generated responses per method, resulting in 1500 responses overall. These results demonstrate that our methods effectively mitigate hallucination while maintaining high generation quality.
>
> Furthermore, we also evaluated our approach using **MM-Vet**, as presented in Table 2. The results confirm that intervening on hallucination heads leads to improved performance in general tasks, further validating the effectiveness of our approach.

---

> ### Author Response · Authors · 2024-11-23
> **Response (Part II)**
>
> **Comment 3**: Some findings look obvious.
>
> a) It is not so surprising that the multi-head-attention is more critical than feed-forward network, since the former takes a majority of parameters and does much more things than the latter in the transformer.
>
> b) It is a well-known fact that hallucination is more related to language bias (rather than) image bias and multimodal models often ignore the information from images compared to that from text.
>
> **Response 3**: Thank you for the discussion. We address parts (a) and (b) separately:
>
> * For part (a): We would like to respectually point out a factual inaccuracy in your claim. In transformers, the MLP modules usually contain more parameters than the multi-head attention (MHA) modules. For instance, in LLaMA-2-7B, MLP modules account for approximately 65% of the total parameters, whereas MHA accounts for only 33% (roughly half of the MLP parameters). Therefore, the assertion that MHAs are more critical solely because they contain more parameters is incorrect.
>
>      * Our findings demonstrate that despite having fewer parameters, MHA plays a disproportionately significant role in affecting hallucination. This insight is new and important, leading deeper investigations into the behavior of MHAs, as detailed in Section 4.2 of our paper.
>
> * For part (b): We assume you are referencing prior works (e.g., Leng et al., 2024; Huang et al., 2024), which have also observed that LVLMs tend to underutilize visual information during text generation. We acknowledge and appreciate the contributions of these studies. However, our work focuses on identifying **specific** network components responsible for hallucination, moving beyond the **broad** notion of language bias.
>
>     * Our findings highlight that not all MHAs are equally implicated in hallucination; rather, a small subset (fewer than 3%) is primarily responsible. Furthermore, we demonstrate that targeted interventions on these components are effective in mitigating hallucination.
> In summary, our work offers new and non-obvious insights into the mechanisms of hallucination in LVLMs and provides actionable strategies to address it. We believe this makes a meaningful contribution to the field.
>
> **Question 4**: A more in-depth analysis is required to explain why the fine-tuning approach is not as effective as the training-free approach.
>
> **Response 4**: We would like to clarify that in our experiments, the fine-tuning approach (TF-HH) performs comparably to the training-free approach (AD-HH). This is evident in Tables 1 and 2. While AD-HH achieves a slightly better average score in Table 1 (24.06 vs. 24.45 for TF-HH), TF-HH slightly outperforms AD-HH in Table 2 (29.9 vs. 29.05).
>
> Theoretically, the training-free approach operates directly in the function space of the Transformer by manipulating the outputs of attention heads (e.g., setting the outputs to zero). In contrast, the fine-tuning approach operates in the parameter space, where the final result is determined by the training method and the model's parameter capacity. This distinction makes it more challenging for the fine-tuning approach to achieve precise manipulations, such as setting outputs to zero, as effectively as the training-free method. However, the fine-tuning approach offers greater flexibility in preserving generation quality by incorporating richer, data-driven adjustments.
>
> We believe both approaches have unique advantages and limitations. In practice, we observe that neither dominates the other, and both achieve comparable performance under different scenarios.
>
> ***
>
> We sincerely thank you for your thoughtful review. We hope the revised manuscript and the clarifications provided above have addressed your concerns effectively. If our responses have satisfactorily resolved your concerns, we would greatly appreciate it if you could consider updating your review score. However, if you have any remaining concerns or require further clarification, please do not hesitate to let us know. We are more than willing to provide additional explanations or updates as needed.

---

### Official Review · Reviewer_pX1o · 2024-11-04

**Soundness:** 3
**Presentation:** 3
**Contribution:** 3
**Rating:** 6
**Confidence:** 3

**Summary:**

The paper observes that some attention heads cause more changes after deactivating them, and name them hallucination heads. There are several characteristics of these heads: (1) they attend more on the images and (2) their parameters move slower than others. Then, the paper proposes one training-free and one training approach to reduce the attention of the hallucination heads on text parts. The experiments on several benchmarks show the benefits of the proposed approach.

**Strengths:**

1. The observation and analysis on the hallucination heads are interesting and well-motivated.

2. The proposed solutions don't require expensive decoding time.

3. The results on several benchmarks look promising.

**Weaknesses:**

1. In Section 4.3, the metrics (CHAIR) and the dataset that is evaluated should be explained.

2. In Tables 1 and 2, it would be better if there is an additional column indicating which method is training-free and which is not.

3. The rest please refer to Questions.

**Questions:**

1. How this method can extend beyond object-level hallucination? For example, do equation (1) and (2) only tell if the existence of the objects but cannot capture if the model incorrectly counts the objects? Then the proposed method can not detect the heads which make hallucination on counting.

2. How many samples is required to compute equation (1)? And what data split do they come from?

3. What's the time required to compute equation (2) to get the scores for all heads?

4. Why the Algorithm 1 directly deactivated the whole text attention weights for the hallucination heads? Isn't it too aggressive as we can see the BLEU scores drop by a lot in Figure 6 (c)? Moreover, why the accuracy on general tasks somehow improves when you apply Algorithm 1? I thought the performance should drop based on the observation in Figure 6 (c).

5. What is the difference between downscaling weight on text attention and upscaling the weight on image attention (Figure 6 (a) and (b))? I thought they meant the same thing as the softmax in attention would keep the sum of text attention and image attention to be one.

---

> ### Author Response · Authors · 2024-11-23
> **Response (Part I)**
>
> Thank you for reviewing our paper and for your positive feedback. We truly appreciate your thoughtful comments and have addressed your concerns and questions below.
>
> **Comment1**: In Section 4.3, the metrics (CHAIR) and the dataset that is evaluated should be explained.
>
> **Response1**: Thanks for this suggestion. We have added more explanation about the CHAIR metrics and te dataset in Section 4.3 to improve readability.
>
> **Comment 2**: In Tables 1 and 2, it would be better if there is an additional column indicating which method is training-free and which is not.
>
> **Response 2**: Thanks for this suggestion. We have added symbol † to denote training-free and symbol * to denote training-based in our paper to make this clear.
>
> **Question 3**:  How this method can extend beyond object-level hallucination? For example, do equation (1) and (2) only tell if the existence of the objects but cannot capture if the model incorrectly counts the objects? Then the proposed method can not detect the heads which make hallucination on counting.
>
> **Response 3**:  Thanks for raising this discussion. We would like to clarify that our framework is flexible and can be extended to attribute errors beyond object-level hallucinations, including errors related to counting, positional reasoning, and other aspects.
>
> Take the hallucination by counting the example you mentioned, we have conducted the experiment on the MME dataset. We take the wrong answer as hallucination token and correct answer as non-hallucination, and identify heads that are mostly associated with counting hallucination applying equation (1) and (2). The counting performance is improved by 5 points from 148 to 153.  Besides, many other aspects such as position, posters, OCR are also improved by adaptively deactivating hallucination heads. Detailed results are reported below:
>
> |             | existence | count | position | color | posters | celebrity | scene | landmark | artwork | OCR |
> |-------------|-----------|-------|----------|-------|---------|-----------|-------|----------|---------|-----|
> | Baseline    | 190       | 148   | 128      | 160   | 139     | 133       | 156   | 162      | 122     | 130 |
> | AD-HH (Ours)| 190       | **153**   |  **138**     | 160   | **142**     | **135**       | **158**   | 162      | 118     | **138** |
>
> **Question 4**: How many samples is required to compute equation (1)? And what data split do they come from?
>
> **Response 4**: We randomly select 1500 samples from the COCO training split to compute equation (1). This detail is provided in the main text (Second paragragh in Section 4.1).
>
> **Question 5**: What is the time required to compute Equation (2) to obtain the scores for all heads?
>
> **Response 5**: Computing Equation (2) for all heads takes about about 1 minutes per sample.  We understand your concerns regarding the computational complexity associated with the number of attention heads. We would like to highlight that there are faster methods for calculating the influence as in Equation (2). These methods leverage techniques like Taylor expansion to approximate the influence of all attention heads in a single backpropagation step (e.g., as described in [1]). Such approaches are highly efficient, requiring only a single forward and backward pass, and the computation time is independent of the number of heads. These approaches can be explored in future work.
>
> [1] Achtibat, Reduan, et al. "Attnlrp: attention-aware layer-wise relevance propagation for transformers." arXiv preprint arXiv:2402.05602 (2024).

---

> > ### Comment · Reviewer_pX1o · 2024-11-24
> >
> > Thank the authors for the reply!
> >
> > Regarding Response 3, thank you for providing the qualitative results here. However, why and what is the intuition that the proposed method can be used for couting? For example, if an image has 2 dogs and the ground truth caption is "the image has two dogs" and the model's prediction is "the image has three dogs", the score in the equation will be 0 because there are no hallucinated objects based on the descriptions in lines 174 - 176.

---

> > > ### Author Response · Authors · 2024-11-25
> > > **Response for Reviewer pX1o**
> > >
> > > We understand your concern. Let us clarify the experiment setting.
> > >
> > > Our experiment setting above is different from what your think. We do not use the extracted object in the caption to apply Equation 1 for this counting task. Instead, we follow the setup in the MME [1], which use prompts like "Are there two dogs on the image? Please answer the question with yes or no."  The model provides a single token "yes" or "no" for response. If the ground truth is "yes" but the model predicts "no". Then, the incorrect token "no" is treated as $y_t$ in Equation (1). In this way, we can continue to apply attribution and intervention methods and observe reduced hallucination rate.
> > >
> > > We acknowledge that your proposal is also applicable. In that case, we can use the number "three" as the hallucination token $y_t$, rather than the object "dog" to calculate Equation (1). In this case, it is more laborious to manually check the hallucination tokens, so we use the setup in MME as a practical example.
> > >
> > > In summary, the formulation of attribution and counterfactual analysis is quite flexible and can be generally applied to tasks beyond object hallucination. Hope this can address your concerns.
> > >
> > > [1] C. Fu et al., MME: A Comprehensive Evaluation Benchmark for Multimodal Large Language Models. Arxiv 2023.

---

> ### Author Response · Authors · 2024-11-23
> **Response (Part II)**
>
> **Question 6**: Why the Algorithm 1 directly deactivated the whole text attention weights for the hallucination heads? Isn't it too aggressive as we can see the BLEU scores drop by a lot in Figure 6 (c)? Moreover, why the accuracy on general tasks somehow improves when you apply Algorithm 1? I thought the performance should drop based on the observation in Figure 6 (c).
>
> **Response 6**: Thank you for highlighting these important points. We would like to clarify a potential misunderstanding: Algorithm 1 employs an input-dependent adaptive strategy to selectively deactivate text attention weights for hallucination heads, unlike the naive strategy used in Figure 6(c), which lacks such adaptability. Therefore, the performance of Algorithm 1 cannot be directly inferred from the observations in Figure 6(c).
>
> To address your concerns about aggressiveness, it is important to note that hallucination heads account for only a small subset (less than 3%) of the Transformer's overall attention mechanism. This ensures that other attention heads remain unaffected, preserving overall generation quality. Thus, it is not overly aggressive to fully deactivate the entire text attention weights for hallucination heads. Evidence for this is provided in Table 4 of the Appendix, which shows that the BLEU score of Algorithm 1 (17.8) is comparable to that of normal generation (17.9).
>
> Finally, the improvement of accuracy in general tasks, as outlined in Table 2, can be attributed to the mitigation of spurious dependencies introduced by hallucination heads. This ultimately enhances model robustness and improves task-specific performance.
>
>
> **Question 7**: What is the difference between downscaling weight on text attention and upscaling the weight on image attention (Figure 6 (a) and (b))? I thought they meant the same thing as the softmax in attention would keep the sum of text attention and image attention to be one.
>
> **Response 7**: In our experiments, scaling is applied *after* computing the softmax attention scores by multiplying a scaling factor to either downscale or upscale specific components. As a result, the sums of text and image attention may not necessarily equal one. This approach allows us to adjust the text attention weights independently without directly affecting the visual information component. Consequently, we can *isolate* and analyze the *separate* contributions of text and visual information more effectively.
>
> ---
>
> We sincerely thank you for your thoughtful review. We hope the revised manuscript and the clarifications provided above have addressed your concerns effectively. If our responses have satisfactorily resolved your concerns, we would greatly appreciate it if you could consider updating your review score. However, if you have any remaining concerns or require further clarification, please do not hesitate to let us know. We are more than willing to provide additional explanations or updates as needed.

---

### Author Response · Authors · 2024-11-23
**General Response and Summary of Changes**

We sincerely thank all reviewers and area chairs for their efforts in reviewing our paper and providing valuable feedback. We have carefully addressed each concern raised by the reviewers and made corresponding revisions to our paper. We highlight the revision in blue. Below, we summarize the key changes in our revision:
- **Section 2 (Page 2)**: We added a discussion of related literature, including [Deng et al., 2024], [Liu et al., 2024], and [Zhang et al., 2024]. Different from these works, our work investigates hallucination through lens of attribution and intervention and designs targeted mitigation strategies.
- **Section 4.3 (Page 6)**: We included descriptions of the metrics (CHAIR) and the dataset used to produce Figure 6, enhancing the figure's readability.
- **Section 5.2 (Table 1,2)**: We added symbols to denote whether methods are training-free or training-based, improving the clarity of the tables.
- **Appendix A.2**:
  - Complete attention maps (Page 16, Figure13): Figure 13 compares the complete attention maps of two typical hallucination and non-hallucination heads across system, image, question, and output tokens. The hallucination head exhibits a clear over-reliance on text tokens. We also add an discussion on the relationship between the text-token over-reliance behaviour and hallucination heads: while hallucination heads frequently show over-reliance on text tokens, not all heads that over-rely on text tokens are hallucination heads.
- **Appendix A.3**:
  - Human-evaluated generation quality (Page 18, Table 6). Table 6 presents the results of a human-based evaluation, focusing on hallucination and generation quality. In this evaluation, our method maintains consistent generation quality comparable to the baseline, while also showing improvements in hallucination reduction.
  - Comparison with additional baselines (Page 18, Table 8). Table 8 provides comparisons with additional three baseline methods, including [Deng et al., 2024], [Liu et al., 2024], and [Zhang et al., 2024]. Although these baselines also achieve a reduction in hallucination rates compared to greedy decoding, our approach, leveraging targeted interventions, demonstrates a greater reduction in hallucinations compared to these baselines.
  - Evaluation on larger and more recent LVLM models (Page 19, Table 9). Table 9 extends the evaluation to larger and more recent models, including Llama-3.2-11B-Vision, LLaVA-v1.5-13B,  Chameleon-30B, and LLaVA-v1.6-34B. Our method achieves up to 10 points in hallucination reduction on these models.
  - Hallucination behaviour across different scales of LVLMs (Page 19, Table 10). Table 10 provides an empirical results on the number and ratio of salient hallucination heads across different scales of LVLMs. The findings indicate that hallucination heads tend to diminish as model size increases and sufficient post-training is applied.

We believe the revisions outlined above have significantly enhanced the quality of our paper, thanks to the insightful feedback from the reviewers. We hope these updated results address the reviewers' concerns and further strengthen the contributions of our work. Thank you once again for your valuable feedback.

---

### Meta-Review · Area_Chair_Zn6P · 2024-12-23

**Metareview:**

This paper investigates hallucination in Large Vision-Language Models (LVLMs) through the lens of causal attribution and intervention. The work identifies specific "hallucination heads" within the multi-head attention mechanism and proposes two methods to mitigate hallucination: a training-free adaptive deactivation approach and a targeted fine-tuning strategy. The paper has solid experiments, and provides a systematic analysis for hallucination in VLMs. The paper also had good responses to questions presented by reviewers. For these reasons I vote to accept this paper!

**Additional Comments On Reviewer Discussion:**

During the rebuttal period, there was extensive discussion between reviewers and authors that substantially strengthened the paper. The initial reviews raised several key concerns: Reviewer pX1o (Score: 6) focused on technical clarifications regarding metrics, implementation details, and methodology, particularly around the CHAIR evaluation and attention weight scaling. Reviewer JmFc (Score: 6) highlighted missing citations, limited evaluation metrics, and the need for deeper analysis of the fine-tuning approach. Reviewer Ex6k (Score: 8) questioned the impact on language generation capabilities and the focus on object hallucination, while Reviewer xVEm (Score: 8) raised concerns about model coverage and technical aspects of hallucination head identification. The authors responded comprehensively to these concerns with substantial additions: they extended their experiments to larger models including LLaVA-34B and Chameleon-30B, conducted human evaluations for generation quality, added comparisons with recent baselines, and provided detailed technical clarifications about their methodology. The authors' thorough response effectively addressed all major concerns while significantly enhancing the paper's contributions through additional experiments and analyses.

---

### Decision · Program_Chairs · 2025-01-22

Accept (Poster)